



# Using Satellite-Based Evapotranspiration Estimates to Improve the Structure of a Simple Conceptual Rainfall-Runoff Model

Tirthankar Roy[1], Hoshin V. Gupta[1], Aleix Serrat-Capdevila[1,2] and Juan B. Valdes[1]

[1] Department of Hydrology and Atmospheric Sciences, The University of Arizona, Tucson, AZ, USA
[2] Water Global Practice, The World Bank, Washington DC, USA

*Correspondence to:* Tirthankar Roy (royt@email.arizona.edu)

**Abstract.** Daily, quasi-global (50°N-S and 180°W-E), satellite-based estimates of actual evapotranspiration at 0.25° spatial resolution have recently become available, generated by the Global Land Evaporation Amsterdam Model (GLEAM). We investigate use of these data to improve the performance of a simple lumped catchment scale hydrologic model driven by satellite-based precipitation estimates to generate streamflow simulations for a poorly gauged basin in Africa. In one approach, we use GLEAM to constrain the evapotranspiration estimates generated by the model, thereby modifying the daily water balance and improving model performance. In an alternative approach, we instead change the structure of the model to improve its ability to simulate actual evapotranspiration (as estimated by GLEAM). Finally, we test whether the GLEAM product is able to further improve the performance of the structurally modified model. The results suggest that the modified model can provide improved simulations of both streamflow and evapotranspiration, even if GLEAM-satellite-based evapotranspiration data are not available.

**Keywords.** Satellite-based actual evapotranspiration, satellite-based precipitation, streamflow simulation, catchment scale modeling, poorly gauged basins, diagnostic model structural improvement, reduction of epistemic uncertainty.

## 1 Introduction

### 1.1 Statement of the Problem

*[1]* As a primary mechanism in the surface-to-atmosphere portion of the water cycle, evapotranspiration (ET) plays a crucial role in the water and energy budgets of a hydrologic system. In practice, ET can be estimated either from model simulations or from remotely sensed observations. For example, ET can be estimated as a residual of water balance computations, or via a land-surface energy budget (e.g. Monteith, 1965; Priestley and Taylor, 1972), and simple empirical physically based schemes (Hargreaves and Samani, 1985) can be applied in data-scarce regions. Ultimately, the quality of a model-derived estimate of ET depends on the various sources of uncertainty (inputs, parameters, process representation, structure, etc.) inherent to the model-based scheme used, and common problems include both over- and under-estimation of evaporative fluxes (Trambauer et al., 2014). Recently, methods that use satellite-based remotely sensed climatic and environmental observations provide an alternate approach to the estimation of ET and its different components (e.g. Bastiaanssen et al., 1998; Arboleda et al., 2005).

*[2]* Several studies have advocated and/or implemented the idea of using physically consistent estimates for the parameters of hydrologic models (Pokhrel et al., 2008, 2012; Savenije, 2010; Schaefli et al., 2011; Kumar et al., 2013; Troch et al., 2015 and references therein). However, in catchment-scale modeling, it is common practice to use parameter estimates that are calibrated by adjusting the simulated streamflows to try and match observed data. If due care is not implemented during the calibration strategy, this approach can result in conceptually unrealistic estimates for the parameters. Such a result defeats an important purpose of using conceptual/physically based models (as opposed to empirical data-based models), which is to help us better understand the dynamical behavior of the system.





*[3]*     In principle, the potential of such models can be better realized by incorporating more information about the physical system during model development. Such information can take various forms and be incorporated in different ways. Evapotranspiration (ET) can be used to constrain model parameters that are sensitive to the ET process (Winsemius et al., 2008; van Emmerik et al., 2015). Alternatively, ET can be used as a calibration target along with streamflow within a multi-objective setting (Zhang et al., 2009). There has also been a recent drive towards structurally flexible models that are able to both better characterize the uncertainty associated with model structure and use additional information to help reduce such uncertainty (Wagener et al., 2001; Marshall et al., 2006; Clark et al., 2008; Savenije, 2010; Schaefli et al., 2011; Fenicia et al., 2008a, 2008b, 2011; Bulygina and Gupta, 2009, 2010, 2011; Martinez and Gupta, 2011; Nearing, 2013; Nearing and Gupta, 2015; Clark et al., 2015).

*[4]*     A variety of satellite-based remotely sensed estimates of daily precipitation have been available for some time (e.g. Hsu et al., 1997; Joyce et al., 2004; Huffman et al., 2007; Funk et al., 2014), making it possible to consider the model-based generation of streamflow simulations for ungaged locations. Recently, satellite-based remotely sensed estimates of daily ET have become available, based on a variety of different retrieval algorithms of varying complexity (e.g. Bastiaanssen et al., 1998; Arboleda et al., 2005; Miralles et al., 2011a). Worldwide evaluations suggest that satellite-based ET estimates are strongly correlated (~0.83) with ground-based observations made at flux towers (Miralles et al., 2015; García et al., 2016). Miralles et al. (2015) compared three process-based ET methods (the Moderate Resolution Imaging Spectroradiometer evaporation product PM-MOD, the Global Land Evaporation Amsterdam Model evaporation product GLEAM, and the Priestley-Taylor Jet Propulsion Laboratory model PT-JPL) against surface water balance from 837 globally distributed catchments, and reported that GLEAM and PT-JPL provide more realistic estimates of ET. They found these two products to provide superior overall performance for most ecosystem and climate regimes, while PM-MOD tends to underestimate the flux in tropics and subtropics.

*[5]*     Of relevance to this study is that the GLEAM product is computed using only a small number of satellite-based inputs. Miralles et al. (2011) have shown that GLEAM estimates of evaporation are strongly correlated (0.80) with annual cumulative evaporation estimated via eddy covariance at 43 stations, and have very low (-5%) average bias. The correlations at individual stations are strong (0.83) for all vegetation and climate conditions, and improve to 0.9 for monthly time series (Miralles et al., 2011).

*[6]*     These reports suggest that satellite-based ET (SET) estimates have the potential to be useful for hydrologic applications. While previous studies have used SET estimates to *constrain the parameters* of hydrologic models (Winsemius et al., 2008; van Emmerik et al., 2015), the recent interest in diagnostic improvements to model structure (Gupta et al., 2008, 2012; Gupta and Nearing, 2014) suggests that it would be potentially more valuable to use the ET data to actually *improve the model structure* when possible. This study attempts to explore this possibility in the context of using satellite-based data to drive a streamflow simulation model for a poorly gauged basin in Africa.

**1.2 Objectives and Scope**

*[7]*     In this study, we explore the use of the GLEAM daily SET product (Miralles et al., 2011; Martens et al., 2016) to improve the performance of a simple lumped catchment scale hydrologic model driven by satellite-based precipitation estimates to generate streamflow simulations for a poorly gauged basin in Africa. We first use the GLEAM product to constrain the evapotranspiration estimates generated by the model, thereby improving the daily water balance. Next, we instead change the structure of the model to improve its ability to simulate actual evapotranspiration (as estimated by GLEAM). Finally, we test whether use of the GLEAM product can further improve the performance of the structurally modified model, to see if further





improvements are achievable. The modified model provides improved simulations of both streamflow and evapotranspiration, even when GLEAM-satellite-based evapotranspiration estimates are no longer available.

## 2 Study Area, Data and Methodology

### 2.1 Study Area

*[8]*    This study is carried out for the Nyangores River Basin (NRB), which is a sub-basin of the Mara River flowing through Kenya (Fig. 1). NRB has an aerial coverage of 697 km² and is located at the northeastern side of the Mara River Basin (MRB; Location: 33°88'E 35°90'E 0°28'S 1°97'S). The perennial Nyangores River originates from the Mau Escarpment (3000 m ASL) fault scarp passing through the western side of the Great Rift Valley in Kenya. It then merges with the Amala River at the Napuiyapi swamp (2932 m ASL) to form the Mara River, which flows all the way to Lake Victoria at Musoma Bay, Tanzania

(1130 m ASL). MRB (or NRB) has two wet seasons consequent to the yearly oscillations of the inter-tropical convergence zone (ITCZ), the primary wet season occurring during March to May (MAM) and the secondary during October to December (OND). The long-term mean rainfall in the Mau Escarpment is around 1500 mm. The rainfall in the basin is influenced by factors like topography, elevation gradient, regional influence of Lake Victoria, sea-surface temperature (SST) of the Indian Ocean, etc.

**[Insert Fig 1]**

### 15   2.2 Data

#### 2.2.1 Estimates of Actual Evapotranspiration

   *[9]*    The source of the SET data used in this study is the Global Land Evaporation Amsterdam Model (GLEAM) Version 3.0. GLEAM comprises a set of algorithms that use remotely sensed climatic and environmental observations to estimate various components of ET. Satellite-based observations of surface net radiation and near-surface air temperature are processed via the

Priestley-Taylor Equation (Priestley and Taylor, 1972) to calculate Potential Evapotranspiration (PET), which is then converted to Actual Evapotranspiration (AET) by incorporating an evaporative stress factor obtained from microwave observations of vegetation optical depth (as a proxy for vegetation water content) and root-zone soil moisture (simulations). Interception loss is calculated using the Gash analytical model (Gash, 1979).

   *[10]*   Three different versions of the GLEAM datasets are currently available, depending on the satellite observations used.

The version used in this study (GLEAM_v3.0b) is based solely on satellite observations, is quasi-global (50°N-S, 180°W-E), has a spatial resolution of 0.25°, and has a daily temporal coverage of 13 years (2003 to 2015).

   *[11]*   Figure 2 shows the annual mean of GLEAM AET (GAET) over the entire MRB. As can be seen, GAET increases towards the western side of the basin. The annual average GAET varies between 900 mm/year to 1200 mm/year. We computed corresponding estimates of PET via the Hargreaves Equation (HPET) using temperature data collected from the six met stations

surrounding the MRB (Fig. 1); maximum and minimum temperatures were averaged to obtain time series of maximum and minimum temperatures and processed through the Hargreaves Equation to calculate PET. For a small number (~0.6%) of days, the lumped GAET values were found to be larger than the lumped HPET values; for these few anomalous values, HPET was replaced by GAET. Figure 3 shows the time series of HPET and GAET for NRB.

**[Insert Fig 2]**

**[Insert Fig 3]**





### 2.2.2 Estimates of Precipitation

*[12]*   The Real Time Multi-satellite Precipitation Analysis (TMPA-RT) of the NASA Tropical Rainfall Measuring Mission combines information from multiple satellites to produce a quasi-global (50°N-S, 180°W-E), near-real-time (March 1, 2000 to near-present) precipitation product at 0.25° × 0.25° spatial and 3-hourly temporal resolution (this product is the real-time version of TMPA (Huffman et al., 2007)). Launched in 1997, TRMM is used primarily for research and development, being the first satellite dedicated to precipitation studies. It is also the first and the only satellite that, in addition to visual, infrared, and passive microwave sensors, also has space borne radar for precipitation measurement using active microwave sensors. The after-real-time TMPA product also incorporates rain gauge information wherever feasible. In this study, we aggregated the 3-hourly TMPA-RT data to daily level, resampled from the coarse resolution (0.25° × 0.25°) to a resolution of 0.05° × 0.05°, and implemented a bias correction using the "Climate Hazards Group InfraRed Precipitation with Station" product (CHIRPS; Funk et al., 2014) and rain gauge measurements (see Roy et al., *in prep*).

### 2.2.3 Estimates of Streamflow

*[13]*   Streamflow data were computed using the calibrated stage-discharge relationship for the Bomet Bridge discharge station (Station ID: 1LA03; Location: 0°47′23.50″S 35°20′47.45″E) on the Nyangores River (drainage area approximately 697 km$^2$), which is one of the two main tributaries of the Mara River. Data is available data for the period Jan 1, 1996 to Jun 30, 2010, during which time only about ~8% of the records are missing.

### 2.2.4 Estimates of Temperature

*[14]*   We computed PET using the Hargreaves Equation (Hargreaves and Samani, 1985), the annual mean of which closely matched the reported PET value for the study area (WREM, 2008). The temperature data used in the Hargreaves Equation were extracted from the Global Surface Summary of the Day (GSOD) product produced by the National Climatic Data Center (NCDC) in Asheville, NC. The daily temperature data includes multiple observations and are available in three forms: maximum, minimum, and average.

### 2.3 The Hydrologic Model

*[15]*   The spatially lumped HyMod Version 1 (HyMod-V1) conceptual rainfall-runoff model with six parameters has previously been implemented for satellite-rainfall-based simulation and forecasting of streamflow for the study area (Roy et al., *in prep*). The model is driven using mean daily precipitation and PET data to generate daily estimates of AET and streamflow. Nonlinear vertical flow processes are controlled by a two-parameter soil moisture accounting (SMA) module based on the Moore (1985) rainfall excess model. Horizontal routing is achieved via a linear (ROUT) module that includes quick-flow routing for fast overland flow and slow-flow routing for baseflow. Details of the model structure and process equations are presented in Appendix A. We will refer to the structurally modified version of the model as HyMod Version 2 (HyMod-V2).

### 2.4 Study Approach

*[16]*   We conducted the investigation in two stages. The first stage consists of five steps designed to improve model performance with respect to both streamflow and evapotranspiration (as assessed against data), but *without* making structural modifications to the model. The strategy includes using GAET to *constrain* simulated evapotranspiration, recalibration of model parameters, and a kind of "bias-correction" of GAET. In doing so, we specifically do *not* directly assimilate GAET into the





model (either by a Bayesian data assimilation "nudging" procedure such as the Ensemble Kalman Filter, or by direct insertion), so that the model's representation of overall water balance is not compromised. Accordingly, while we _are_ extracting information from the GLEAM product, we do so via a process of "*constraining*" rather than "*assimilation*".

*[17]*  In the second stage, we modify the structure of the model to directly improve its ability to simulate ET (using GAET as
the target). The steps followed in stage one are repeated so that results of the different strategies can be compared.

*[18]*  Conceptually, the difference between Stages I and II is that, in the former the information provided by GAET is used only to constrain the evapotranspiration fluxes and soil moisture states of the (re-calibrated) model, whereas in the latter the information contained in GAET is used to alter the model structure. While the former provides a temporary improvement to model performance, achieved as long as GLEAM data are available, the latter is expected to provide a lasting improvement to
model performance that should persist even when GLEAM data are not available. In the final step of Stage II, we check to see whether the GAET product contains residual information that, not having yet been used to improve the model structure, remains useful for improving model performance via the constraining operation.

### 2.4.1 Stage One – Constraining HyMod using GAET

**Step I-1:**   **Benchmark:** HyMod-V1 is forced with TMPA-RT satellite-based precipitation and the parameters are calibrated
against observed streamflows. GAET satellite-based evapotranspiration is not used. This is the benchmark step against which model performance improvements are assessed.

**Step I-2:**   **GAET Constrained:** GAET is used to constrain HyMod-V1 HAET. The model parameters remain the same as in Step I.

**Step I-3:**   **Recalibrated GAET Constrained:** GAET is used to constrain HyMod-V1 HAET. The model parameters are re-
calibrated to match simulated to observed streamflows.

**Step I-4:**   **Bias-Corrected Recalibrated GAET Constrained:** A "bias-correction" is applied to GAET, and the bias-correction and model parameters are calibrated together to match simulated and observed streamflows. Since no ground-based data are available to bias-correct GAET, we tested two empirical bias correction schemes (shown below):

$$y = f(x, \cdots) \text{ and } x = \boldsymbol{a} \cdot \text{GAET} \tag{1}$$

$$y = f(x, \cdots) \text{ and } x = \boldsymbol{a} \cdot \text{GAET}^{\boldsymbol{b}} \tag{2}$$

where **y** represents the streamflows after bias correction of GAET, and $\boldsymbol{a}$ and $\boldsymbol{b}$ are parameters of the bias-correction formulation ($\boldsymbol{a}$ controls the variance and $\boldsymbol{b}$ controls the degree of non-linearity).

**Step I-5:**   **GAET Removed:** The model obtained in Step IV is run without using GAET as a constraint. No re-calibration is performed. This shows what would happen to model performance if real-time GAET data were to become unavailable.





**2.4.2 Stage Two – Modifying the Structure of HyMod using GAET**

**Step II-1:** **Structurally Modified:** The evapotranspiration equation of HyMod-V1 is modified to improve its ability to reproduce GAET (i.e., so that HAET becomes more similar to GAET). Four evapotranspiration equations of progressive complexity are tested. In each case the model parameters are re-calibrated to match simulated streamflows to observed data. The final result is a structurally modified model called HyMod-V2.

**Step II-2:** **GAET Constrained:** Using the "best" structurally modified model (HyMod-V2) from Step II-1, we repeat Step I-2.

**Step II-3:** **Recalibrated GAET Constrained:** GAET is used to constrain HAET simulated by HyMod-V2. The model parameters are re-calibrated to match simulated and observed streamflows.

**Step II-4:** **Bias-Corrected Recalibrated GAET Constrained:** The empirical "bias-correction" scheme is applied to GAET, and the bias-correction and modified-model (HyMod-V2) parameters are calibrated together to match simulated and observed streamflows.

**Step II-5:** **GAET Removed:** HyMod-V2 is run without using GAET as a constraint. No re-calibration is performed. This shows what would happen to performance of HyMod-V2 if real-time GAET data were to become unavailable.

**2.5 Calibration Methodology and Benchmark Model Calibration**

*[19]* Calibration of the model (and bias-correction) parameters was performed by running the SCE-UA algorithm (Duan et al., 1992) with 10 complexes for 25 loops. Calibration was performed to match streamflows in a $\lambda$-transformed space (Box and Cox, 1964; see Appendix B) to minimize the effects of skewness and reduce heteroscedasticity. The performance criterion used was the Mean Squared Error (MSE) of transformed flows. The model was run continuously for the 7.5-year period Jan 2003 to June 2010, with the first 4 years (2003 to 2006) used for calibration and the remaining 3.5 years (2007 to mid-2010) used to provide an additional assessment of model performance. Results are shown for the "*calibration (4-years)*", "*evaluation (3.5 years)*" and "*total (7.5 years)*" simulation periods. All the streamflow errors statistics reported in this study are in the $\lambda$-transformed space.

**2.6 Metrics Used for Performance Evaluation**

*[20]* Four metrics are used in this study to assess the model performance (Table 1). These metrics measure performance in regards to overall mean squared error, bias, variability, and correlation (see Gupta et al., 2009), are computed in the transformed space where applicable (e.g. for streamflows), and are normalized to be comparable.

**[Insert Table 1]**

**3 Results**

**3.1 Results from Stage I (Constraining Simulated AET)**

**3.1.1 Benchmark Model (Step I-1)**

*[21]* The performance of the benchmark model HyMod-V1, driven using TMPA-RT satellite-based precipitation and with parameters calibrated to match simulated streamflow to observed data, is reported in Table 4. The NMSE varies between 0.56





(calibration period) and 0.84 (evaluation period), where NMSE = 0.56 means that on average only about 44% (1.0 - 0.56 = 0.44) of the variability in the flows has been explained. This is not surprising given the use of a simple lumped conceptual model driven by satellite-based estimates of precipitation for a poorly gauged basin. The flow biases are small (NBμ < 15%) indicating that long-term water balance is approximately preserved. The calibrated values of the model parameters are reported in Appendix

C.

*[22]*      Table 2 presents a comparison of the model generated HAET with the GAET data (for the total 7.5 year simulation period). HAET tends to be larger on average, varies over a wider range, is considerably more skewed, and is less kurtotic. Some of the reasons for this can be understood from the time-series plot and scatterplot shown in Fig. 4. The behavior of HAET tends to be more erratic and, although both HAET and GAET show seasonal patterns, the former regularly drops to zero or near zero

(explained by the very simple, threshold-like, ET process representation in the model, which does not contain a resistance term). The result is that HAET and GAET are not well correlated (Fig. 4b) and have different shapes for their empirical probability distributions (Fig. 5). Even if we were to ignore the time-steps when HAET drops closer to zero, HAET is strongly positively biased (too large), which results from trying to satisfy potential evapotranspiration (PET).

*[23]*      Table 3 reports a water balance estimate WBAET of the mean annual AET for the basin, obtained by subtracting mean

annual streamflow (at the discharge station) from mean annual precipitation (estimated from TMPA-RT). WBAET is similar in magnitude to GAET, and we have GAET < WBAET < HAET < HPET, indicating that the AET computed by the model tends to be a little high.

**[Insert Table 2]**

**[Insert Fig 4]**

**[Insert Fig 5]**

**[Insert Table 3]**

### 3.1.2 Using GLEAM AET to Constrain the Model (Step I-2)

*[24]*      Next, GLEAM satellite-based daily estimates of AET (GAET) were used to constrain the HAET estimates generated by HyMod-V1. The constraint is imposed by modifying the original evapotranspiration equation of the model (Eq. A5) from

HAET = $\min\{\text{PET}, C_{SMA}\}$ to the new form HAET = $\min\{\text{PET}, \text{GAET}, C_{SMA}\}$; this is not a 'structural' modification to the form of the process equation – it simply constrains HAET ≤ GAET. The practical effect of this modification is that daily changes in soil moisture storage become smoother and thereby closer to our expectation of how they should behave. The parameters of the model were *not* recalibrated.

*[25]*      Table 4 indicates that the model performance has become significantly worse due to streamflow becoming positively

biased. Given that GAET < HAET on average in the previous step, this makes sense because imposing GAET as a constraint alters the water balance of the model.

### 3.1.3 Recalibration of the Model Constrained Using GLEAM AET (Step I-3)

*[26]*      To try and fix the water-balance problem introduced during Step I-2, we recalibrated the parameters of the model to improve the match to observed streamflows (while continuing to use GAET to constrain HAET in the model). Although the large

positive bias was reduced (Table 4) and the NMSE statistic is improved compared to Step I-2, most of the error statistics deteriorated for all three periods (calibration, evaluation, and total simulation) compared to Step I-1. Importantly, this calibration



step resulted in an *unrealistically large* value of 17.36 meters for the size of the soil moisture storage (previously a more realistic 0.76 meters). This value is clearly conceptually and physically inconsistent (the realistic range is about zero to 2 meters), and while it improves calibration period performance, the lack of consistency is reflected in the sharp deterioration in performance during evaluation. Unrealistic parameter values, such as this, are indications of either severe errors in the data, or structural errors in the model. Since GAET agrees well with WBAET on average, it is likely that the major cause here is model structural inadequacy (Gupta et al., 2012). For completeness, we next check (Step I-4) to see whether this problem can instead be resolved by implementing an empirical bias-correction to GAET.

### 3.1.4 Applying a Bias Correction to the GLEAM AET (Step I-4)

*[27]* We tested two empirical bias correction schemes (Eq. 1 and 2) applied to the GAET data to obtain a quantity we call BC-GAET, and calibrated the additional parameters (from these equations) along with the HyMod-V1 parameters. Results for both schemes were similar, but Eq. 2 provided slightly better performance for the evaluation and total simulation periods and so we selected Eq. 2. Compared to the Benchmark Step I-1, NMSE and NB$\mu$ calibration period statistics reduced from 0.56 to 0.43 and 12% to 6%, respectively (Table 4) while $\rho$ increased from (0.76/0.66/0.72; Cal/Eval/Tot) to (0.83/0.74/0.78). Perhaps more important, the calibrated value of parameter H is now 0.65 meters, which is within the conceptually acceptable range.

### 3.1.5 GLEAM Data Removed (Step I-5)

*[28]* Finally, the model obtained in Step I-4 was run without the use of GAET (actually BC-GAET) to see how well the model would perform if GAET data were to become unavailable. The results (Table 4) results indicate that model performance does not deteriorate significantly when GAET data become unavailable and, in some cases, is better than the benchmark.

**[Insert Table 4]**

### 3.2 Results from Stage II (Modifying Model Structure)

### 3.2.1 Modifying the Model Structure to Improve Simulation of AET (as Estimated by GLEAM) (Step II-1)

*[29]* Results from Stage I confirm that assimilating GAET into HyMod can improve the overall model performance. However, for operational implementation, the method requires real-time estimates of SET, which could sometimes pose a challenge for practical application. To overcome the need for real-time data availability a simple approach could be to establish a functional relationship between HAET and GAET from historical records and use that relationship to adjust HAET. In our case, however, HAET and GAET did not show a sufficiently strong relationship (Fig. 4). Therefore, we instead investigated whether we could use the historical GAET data to improve the structure of the model itself.

*[30]* Our previous results (Fig. 4) showed that HAET generated by HyMod-V1 did not match well with GAET. This is likely because the entire soil moisture storage is exposed to the ET process and there is representation of the physical processes that act to inhibit ET when the soil moisture content is low. Consequently, it is common for all of the soil moisture to evaporate away during a single time step, leaving no water available for evaporation at the next time step (provided no precipitation is added), so that HAET drops to zero. This tendency can be reduced by modifying the ET process representation so that HAET more closely follows GAET. We do this by multiplying the ET equation by a function $K(\cdot)$ such that $0 \le K(\cdot) \le 1$.




[31]     We tested four different forms for $K(\cdot)$ that represent incremental increases in complexity (Table 5). Writing the main ET equation in the general form:

$$Y_t = K_t \cdot X_t \cdot EDR_t \tag{3}$$

where $Y_t$ is the model generated AET ($HAET_t$), $X_t$ is the soil moisture storage ($C_{SMA_t}$), and $EDR_t$ is the *evaporation demand ratio* computed as $\min\left\{1, \frac{PET_t}{X_t}\right\}$. The most general form for $K_t$ is given by:

$$K_t = K_{min} + [K_{max} - K_{min}] \cdot f(\psi_t) \tag{4}$$

where, $K_{min}$ and $K_{max}$ are lower and the upper limits for K, and $\psi_t$ is the ratio of actual to maximum storage capacity ($\psi_t = X_t/X_{max}$).

**[Insert Table 5]**

### (a) Results from Step II-1a

[32]     This step is identical to the benchmark step (Sect. 2.5) where the calibrated HyMod-V1 is run without GAET estimates. Results are summarized in Table 6.

### (b) Results from Step II-1b

[33]     In this case, $K_t = K_0$ is applied as a constant multiplier to the ET equation (Table 5), thereby acting as a constant surface resistance to ET. Calibration (of all of the model parameters) resulted in improved error statistics (Table 6). The estimate obtained for the surface resistance was $K_0 = 0.73$. However, we again obtained a conceptually unrealistic value (H = 9.5 meters) for the soil moisture storage parameter.

### (c) Results from Step II-1c

[34]     In this case, the more complex form $K_t = K_0 \cdot f(\psi_t)$ was used (Table 5). This produced a model performance (Table 6) comparable to that of the previous Step II-1b, but with a more realistic value for the calibrated value of soil moisture storage (H = 0.90 meters). Interestingly, the calibrated value for $K_0$ was 1, implying that $K_0$ becomes irrelevant once $f(\psi_t)$ is introduced to the ET equation.

### (d) Results from Step II-1d

[35]     Finally, the most complex form $K_t = K_{min} + [K_{max} - K_{min}] \cdot f(\psi_t)$ was used. The calibration parameter $K_0$ was used to represent $K_{max}$, and $K_{min}$ was defined as $K_{min} = \gamma \cdot K_{max}$ via a second calibration parameter $\gamma$ (ranging from 0 to 1) (Table 5). Results indicate that although the calibration error statistics (Table 6) are similar to that of Step II-1c, the evaluation and total simulation statistics are better. The calibrated value of parameter 'BE' (derived by transforming the parameter 'be'; see Eq. B1) was 0.86, indicating only a mildly non-linear relationship between $\psi_t$ and K (or HAET). The minimum and the maximum limits




of $K_0$ were close to zero and one, respectively, confirming the findings of Step II-1c that once $f(\psi_t)$ is introduced into the ET equation, the need for the $K_{max}$, and $K_{min}$ parameters largely disappears.

**[Insert Table 6]**

### 3.2.2 Final Model Selection from Step II-1

*[36]* In this section, we address two main questions: (a) Does the model structural modification (to the representation of the ET process) improve ET estimation, and (b) if so then what level of complexity is adequate? Table 7 presents the streamflow and AET performance statistics for the total simulation period for the four cases. Since Step II-1a provided very poor error statistics for AET (e.g. NMSE = 8.93 and NBσ = 1.89), we disregarded this case. Although Step II-1b provided the best NBσ (-0.06) statistics for streamflow, and the best NMSE (1.28) and NBμ (0.12) statistics for AET, the value obtained for the soil moisture storage capacity (H) was unrealistic; we therefore also disregarded Step II-1b. Comparison of Step II-1c and Step II-1d shows that while their streamflow and AET simulations were similar (Fig. 6), Step II-1d provided slightly better NMSE (0.70) and NBμ (0.13) statistics for streamflow and slightly better R (0.49) statistics for AET (Table 7 and Fig. 7). We therefore selected the most complex form $K_t = K_{min} + [K_{max} - K_{min}] \cdot f(\psi_t)$ for the ET function (Step II-1d). The corresponding model is hereafter referred to as 'HyMod-V2'.

**[Insert Table 7]**

**[Insert Fig 6]**

**[Insert Fig 7]**

*[37]* Comparing the streamflow error statistics of Step I-4 (Table 4) and Step II-1d (Table 6), we see that they are quite similar, indicating that the ET constraining (first approach) and diagnostic structural improvement (second approach) strategies produce dynamical behaviors that are similar (as measured by the four performance metrics used).

*[38]* The modified model (HyMod-V2) was next used with GAET in a similar manner to Steps I-2 to I-5, to address two questions; i) could more information from GAET be assimilated into the model, or is the improved model structure (without GAET) already good enough, and ii) is the bias adjustment of GAET (Step-IV) necessary once the model structure has been improved?

### 3.2.3 Using GLEAM AET to Constrain the Modified Model (Step II-2)

*[39]* GAET was used to constrain the ET process in HyMod-V2, without model recalibration (the parameters used were from Step II-1d). This introduced significant overestimation bias in the simulation of streamflows (Table 8). Clearly, recalibration of the modified model is necessary when GAET is imposed as a constraint on ET.

### 3.2.4 Recalibration of the Modified Model Constrained Using GLEAM AET (Step II-3)

*[40]* Recalibration of HyMod-V2 improved the error statistics (Table 8); compare these results with the Step I-3 results in Table 4 derived the same way for HyMod-V1. While a small improvement was obtained for the soil moisture storage capacity parameter H (reduced from 17.4 meters to 12.8 meters), this value remained conceptually inconsistent (too large). Overall, the error statistics deteriorated compared to the best results from Step II-1.



### 3.2.5 Applying a Bias Correction to the GLEAM AET (Step II-4)

*[41]*     The HyMod-V2 parameters were calibrated along with the parameters of the GAET bias correction equation (using Eq. 2), as in Step I-4. Although the results improved (Table 8), and the value of H parameter became conceptually realistic (0.88

meters), the results were not significantly different from Step II-1.

### 3.2.6 GLEAM Data Removed (Step II-5)

*[42]*     This step is similar to Step I-5 but with HyMod-V2. The performance of the model remained stable even when GAET was not used (Table 8).

**[Insert Table 8]**

*[43]*     In regards to the two questions that motivated this section, the results indicate that; (a) once information from GAET was assimilated into the model as a modification to the structure there was no further need for use of GAET to constrain the simulation of ET (use of GAET even caused some of the results to deteriorate), and (b) implementation of a bias correction to GAET (Step II-4) did improve the error statistics and resulted in a more conceptually realistic value for the H parameter in the modified model.

### 3.3 Overall Comparison and Analysis of Uncertainty (in Streamflow and AET)

*[44]*     Figure 8 compares the streamflow time series obtained from Step I-4 (constraining ET) and Step II-1d (structural modification) against the benchmark (Step I-1) in both actual and λ-transformed space. Simulations from the structurally modified model HyMod-V2 (Step II-1d) follow the observations most closely, followed by the simulations from Step I-4 (ET constraining) and Step I-1 (Benchmark). Clearly, while the streamflow simulations are improved by both ET constraining and

model structural modification, the latter performs best.

**[Insert Fig 8]**

*[45]*     Using the best model from Step II (HyMod-V2), we next investigate the change in simulation uncertainties for streamflow and AET due to the model structural improvement. Following Roy et al. (*in prep*), we computed the calibration period residual distributions (assumed stationary) in the λ-transformed space and superimposed them on the daily estimates of

the corresponding variables for the total simulation period. Figure 9 shows the histograms of calibration period residuals for the Benchmark and Final steps (Step II-1d). In both cases (AET and streamflow) the residuals become more normally distributed, with the improvement being more prominent for AET. This result is expected, since HyMod-V1 in Step-I showed poor performance in regards to AET. Overall, the structural modification is clearly beneficial.

**[Insert Fig 9]**

*[46]*     Figure 10 shows the streamflow and AET time series along with their corresponding 90% confidence intervals for the Benchmark and the Final steps. Both the streamflow and AET simulations improve as a result of the model structural modification. Although the streamflow uncertainty bounds have not narrowed significantly, the flow series is clearly less biased and tracks the recessions better. Meanwhile the AET simulations have improved significantly: (a) the bias has been reduced, (b)




the uncertainty bounds are narrower, and (c) the erratic behavior originally seen in the AET simulations (frequent drops to zero) has disappeared. Further, although the improvement in streamflow performance is evident from the statistics in Table 4 and 6, the improved behavior is even more apparent in Figure 10 where the model can be seen to now track the recessions quite well.

**[Insert Fig 10]**

**4 Discussion and Conclusions**

*[47]*       This study has explored two different approaches to the use of recently available SET data from GLEAM to improve the realism and performance of the conceptual catchment-scale hydrologic model HyMod. In the first approach, SET data were used to constrain the ET estimates, while in the second we modified the model structure itself. Our study shows that use of satellite-based information can clearly benefit the process of hydrologic modeling for poorly gauged basins by providing new sources of
information to reduce the epistemic component of model structural uncertainty through improved physical process representation.

**4.1 Constraining ET**

*[48]*       Use of ET data to constrain model simulations can improve streamflow forecasts, provided some additional processing steps are implemented. Direct insertion of GAET into the ET equation of the HyMod model resulted in bias (Step I-2); the type
of bias will, of course, be subject to change depending on the SET data used. While recalibration of the model improved model performance (Step I-3), it resulted in a conceptually unrealistic estimate for the storage capacity of the basin. Application of a bias correction to GAET both improved the streamflow forecasts, and resulted in a more realistic estimate for basin storage capacity (Step I-4); use of multiplicative and power-law type bias correction schemes produced almost similar results, with slightly better error statistics for the latter.

*[49]*       The ET constraining approach can be implemented for real-time forecasting, provided that real-time SET estimates are available. Once the HyMod model structure was appropriately modified to provide good simulations, the model simulations were robust/stable, and the streamflow forecasts did not deteriorate when GAET data was removed. Recalibrating the model (against streamflow) when implemented with GAET assimilation did not improve the model performance when the model was run without GAET (Step I-5). Our results suggest, therefore, the ET constraining approach be implemented only for simulation
periods when SET data are available.

**4.2 Structural Modification**

*[50]*       SET data can prove useful for improving the representation of the ET process in a hydrologic model. Our study was able to derive a simple structural form for HyMod that is robust and enabled the model to produce more accurate estimates of AET. We tested several conceptually reasonable structural modifications to the model of varying levels of complexity (Step IIa-
IId), and selected the one that provided the best simulations of both GAET and observed streamflow.

*[51]*       We found that relatively simple changes to the HyMod ET equation significantly improved the ET simulations (as assessed by GAET). However, while a simple multiplicative factor (parameter K) to control AET produced excellent streamflow and AET forecasts (Step II-1b), it resulted in an unrealistic estimate of basin storage capacity. In contrast, inclusion of a soil moisture dependent function $f(\psi_t)$, representing the resistance to evaporation, resulted in a more realistic estimate of basin





storage capacity without compromising the streamflow and AET simulations. The final model structure (HyMod-V2) establishes a non-linear relationship between AET and evaporative demand.

[52]     Overall, the modified model structure provided significantly improved AET forecasts with much narrower uncertainty intervals (see Fig. 10), along with reduced bias in streamflow and improved tracking of the streamflow recessions.

## 4.3 Overall Outlook

[53]     The validation and total simulation streamflow error statistics from Step I-4 (Table 4) and Step II-1d (Table 6) were quite similar, indicating that both the ET constraining (first approach) and diagnostic structural improvement (second approach) can produce comparable results. However, the latter also resulted in improved AET forecasts, even when SET data were made unavailable. Combining the two schemes did not result in notable additional improvements.

[54]     Overall, by incorporating an additional source of external information in a sensible manner (here by structural modification), the need for calibration can be reduced (note that the model was not calibrated against GAET); see the extensive discussion by Gharari et al., (2014) and Bahremand (2016) on this topic. Nevertheless, given the simplistic nature of the hydrologic models and the large uncertainties that exist therein, some degree of calibration will generally remain important and relevant. We do not mean to imply, therefore that calibration is not essential, because we will rarely (correctly) know everything we need to know about the system we are modeling. Instead, we should be aware of the strengths and weaknesses involved in the use of calibration and apply it carefully in such a way that useful information is gained about the underlying nature of the actual physical system. In this study, we demonstrate the need for both approaches. On the one hand, improving the model structure resulted in improved AET simulations without any need for calibration (to AET). On the other hand, the best streamflow performance was achieved when the modified model structure modification was tuned via a calibration procedure.

[55]     Note that this study is based on testing of a single catchment scale conceptual rainfall-runoff model on a single basin, using a single satellite-based precipitation product and a single satellite-based AET product. While not demonstrating universal applicability, the results are clearly indicative and the methodology illustrates how such data can be used to investigate potential improvements to the structures of simple catchment scale models used for hydrologic studies in data scarce regions. For more detailed process-based models, the ET process parameters can be calibrated against some reliable SET estimates (e.g. GLEAM), or the process representation itself can be improved by adapting some similar strategies the SET products are based on.

## 4.4 Conclusions

[56]     In conclusion, SET data can be used to improve model performance in different ways. However, data assimilation strategies that result in model structural modifications can generally be expected to provide longer lasting benefits than ones that simply update or constrain the state trajectories of the model. This is because structural modifications can both improve the initial estimates of the state at each time step, and sustain these improvements into future time steps (Bulygina and Gupta, 2009, 2010, 2011; Nearing and Gupta, 2015). In contrast, even though data assimilation to directly improve state estimates can improve model performance, inadequacies in model structure will tend to cause the state estimates to drift away from their ideal values over time, so that performance deteriorates markedly when the constraining data are not available. Of course, we have only tested a '*constraining*' strategy to assimilating ET information, which is a relatively simple form of data assimilation (DA) (Houser et al., 1998), and more sophisticated approaches such as the Ensemble Kalman Filter (EnKF) could instead be





implemented. However, the efficiency of the EnKF for soil moisture retrieval has been shown to be as low as 30% Nearing et al. (2013a, 2013b) and so it is not clear that more sophisticated forms of DA are justified, especially given the large uncertainties associated with both the data and the model structure for this poorly gauged catchment. We leave such investigation for future work.

## 5   Code and Data Availability

Data and codes (HyMod-V2 in Matlab) used in this study are available on request from the corresponding author, Tirthankar Roy (royt@email.arizona.edu).

## Conflicts of Interests

The authors express no conflicts of interests.

## 10   Acknowledgements

This study was supported by the NASA-USAID SERVIR Program through the award 11-SERVIR11-58. The second author acknowledges support by the Australian Centre of Excellence for Climate System Science (CE110001028), and from the EU-funded project ''Sustainable Water Action (SWAN): Building Research Links Between EU and US'' (INCO-20011-7.6 grant 294947).

30

35





# APPENDIX

## *Appendix A: Original HyMod Equations*

The benchmark version of the spatially lumped conceptual rainfall-runoff model HyMod has six parameters. The model is driven by mean daily precipitation and PET data to generate daily estimates of AET and streamflow. It has two main components, a two-parameter soil moisture accounting (SMA) module based on the Moore (1985) rainfall excess concept, and a linear routing (ROUT) module with parallel quick-flow (fast overland flow) and slow-flow (baseflow) pathways. In the SMA module, the *state variable* (soil moisture storage, C) and the *indicator variable* (storage height, H) are non-linearly related via the following equation (Moore, 1985):

$$C(t) = C_{max}\left(1 - \left(1 - \frac{H(t)}{H_{max}}\right)^{1+b}\right) \tag{A1}$$

where the *maximum storage capacity* ($C_{max}$) and the maximum indicator height ($H_{max}$) are related as:

$$C_{max} = \frac{H_{max}}{1+b} \tag{A2}$$

First, the *initial storage* ($C_{beg}$) is calculated from the initial indicator height ($H_{beg}$) using Eq. A1. Next, $H_{max}$ is subtracted from the sum of precipitation (P) and $H_{beg}$ to calculate *overland flow* (OV) as:

$$OV = P + H_{beg} - H_{max} \tag{A3}$$

*Infiltration* (I) is then calculated by subtracting OV from P:

$$I = P - OV \tag{A4}$$

and an intermediate indicator height ($H_{int}$) is computed by adding I to $H_{beg}$, and used to calculate the *intermediate storage* ($C_{int}$) via Eq. A2. By subtracting $C_{int}$ from the sum of I and $C_{beg}$ we obtain the *interflow* (IF). Finally, the *total runoff* is obtained by adding together OV and IF.

Finally, the HyMod AET (called HAET) is taken to be the smaller of available water $C_{int}$ and potential demand PET (which is provided as input to the model):

$$HAET = \min\{PET, C_{int}\} \tag{A5}$$

and the storage at the end of the time step is computed by subtracting AET from $C_{int}$:

$$C_{end} = C_{int} - HAET \tag{A6}$$

The power coefficients in HyMod ('BE' in Table 5 and 'b' in Eq. A1 & A2) can have values ranging from 0 to infinity. For calibration it is useful to be able to impose finite values to the feasible ranges of the parameters; therefore we applied the following transformation (Eq. A7) which converts the [0,inf) range of parameter BE to the [0,2) range of transformed parameter 'be' so that the search can be conducted on finite range of parameter 'be' instead (similarly for parameter 'b' in Eq. A1 and Eq. A2):

$$BE = \ln(1 - be/2)/\ln(0.5); \quad be = [0, 2) \tag{A7}$$





*Appendix B: The λ-Transformation Used*

The λ-transformation on streamflows used in this work is given by the equation:

$$TQ_t = \left(\frac{Q_t}{\mu_{Qobs}}\right)^{\lambda} \tag{B1}$$

where $Q_t$ and $TQ_t$ represent streamflows in the actual space and the transformed space, $\mu_{Qobs}$ is the mean of the observations in the actual space, and λ is the transformation parameter that reduces the skewness. This expression differs slightly from the form

5    $TQ_t = \frac{(Q_t)^{\lambda}-1}{\lambda}$ recommended by (Box and Cox, 1964), in that the flows are normalized by the mean $\mu_{Qobs}$ instead of by 1.0 before transformation, and the transformed flows all remain positive. This form works as long as the transformation parameter $\lambda \neq 0$ which is true in our case; if $\lambda = 0$, then one should use $TQ_t = \ln(Q_t)$ as discussed by Box and Cox (1964).

*Appendix C: Calibrated HyMod (Actual and Modified) Parameters*

The following table provides calibrated parameters of the actual and the modified HyMod models.

| Para | Step I-1 | Step I-3 | Step I-4 | Step II-1b | Step II-1c | Step II-1d | Step II-3 | Step II-4 |
|------|------|------|------|------|------|------|------|------|
| H | 761.0 | 17364.0 | 646.4 | 9494.0 | 903.7 | 866.0 | 12763.8 | 878.7 |
| B | 1.93 | 1.95 | 1.24 | 1.87 | 0.29 | 0.34 | 1.86 | 0.32 |
| α | 0.48 | 0.37 | 0.67 | 0.38 | 0.31 | 0.27 | 0.45 | 0.31 |
| Nq | 1.44 | 4.54 | 1.25 | 4.22 | 4.80 | 4.71 | 3.51 | 4.50 |
| Ks | 0.00 | 0.09 | 0.00 | 0.10 | 0.10 | 0.09 | 0.10 | 0.10 |
| Kq | 0.10 | 0.22 | 0.10 | 0.19 | 0.24 | 0.24 | 0.18 | 0.20 |
| Kmax | - | - | - | 0.73 | 1.00 | 1.00 | 1.00 | 1.00 |
| γ | - | - | - | - | - | 0.00 | 1.00 | 0.00 |
| BE | - | - | - | - | - | 0.86 | 0.06 | 0.90 |
| a | - | - | 1.33 | - | - | - | - | 1.82 |
| b | - | - | 0.93 | - | - | - | - | 2.00 |



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





# TABLES

Table 1: Performance evaluation metrics used in this study.

| Metrics | Equations |
|---|---|
| Normalized Mean Square Error (NMSE) | $\text{MSE} = \text{mean}((O_i - S_i)^2); \ \text{NMSE} = \frac{\text{MSE}}{\text{var}(O)}$ |
| Normalized Bias in Mean (NBμ) | $\text{NB}\mu = \dfrac{\text{mean}(S) - \text{mean}(O)}{\text{mean}(O)}$ |
| Normalized Bias in Standard Deviation (NBσ) | $\text{NB}\sigma = \dfrac{\text{std}(S) - \text{std}(O)}{\text{std}(O)}$ |
| Correlation Coefficient (ρ) | $\rho = \dfrac{\sum_{i=1}^{N}(O_i - \text{mean}(O))(S_i - \text{mean}(S))}{\sqrt{\sum_{i=1}^{N}(O_i - \text{mean}(O))^2 \sum_{i=1}^{N}(S_i - \text{mean}(S))^2}}$ |
| O: Observed flows; S: Simulated flows; N: Number of data points. | |

Table 2:  Descriptive statistics of GAET and HAET.

| Statistics | GAET | HAET |
|---|---|---|
| Maximum | 4.62 | 6.12 |
| Minimum | 0.119 | 0.00 |
| Mean | 3.03 | 3.52 |
| Median | 3.08 | 4.21 |
| Mode | 0.11 | 0.00 |
| Std. Dev. | 0.59 | 1.72 |
| Skewness | -0.58 | -1.03 |
| Kurtosis | 3.84 | 2.62 |

Table 3: Annual mean of AETs and HPET.

| Source | Annual Mean (mm) |
|---|---|
| GAET | 1100 |
| HAET | 1263 |
| WBAET | 1146 |
| HPET | 1704 |





Table 4: Streamflow error statistics for calibration, evaluation, and total simulation (in parenthesis) for all five different cases in Stage I analysis.

| Calibration | | | | | |
|---|---|---|---|---|---|
| **Metrics** | **Step I-1** | **Step I-2** | **Step I-3** | **Step I-4** | **Step I-5** |
| **NMSE** | 0.56 | 1.68 | 0.64 | 0.43 | 0.60 |
| **NBμ** | 0.09 | 0.32 | 0.12 | 0.09 | -0.09 |
| **NBσ** | -0.12 | -0.15 | -0.19 | -0.06 | -0.04 |
| **R** | 0.76 | 0.81 | 0.74 | 0.83 | 0.75 |
| Evaluation (Total Simulation) | | | | | |
| **Metrics** | **Step I-1** | **Step I-2** | **Step I-3** | **Step I-4** | **Step I-5** |
| **NMSE** | 0.84 (0.77) | 2.13 (2.17) | 0.92 (1.19) | 0.88 (0.75) | 0.64 (0.56) |
| **NBμ** | 0.14 (0.14) | 0.38 (0.38) | 0.09 (0.22) | 0.15 (0.16) | -0.01 (-0.02) |
| **NBσ** | -0.04 (-0.08) | -0.03 (-0.10) | 0.04 (-0.01) | 0.17 (0.02) | 0.14 (0.05) |
| **R** | 0.66 (0.72) | 0.71 (0.76) | 0.61 (0.69) | 0.74 (0.78) | 0.73 (0.74) |

Table 5: K-function in different cases.

| Cases | $K_{max}$ | $K_{min}$ | $f(\psi_t)$ | Additional Parameters |
|---|---|---|---|---|
| a | 1 | 0 | 1 | None |
| b | $K_0$ | 0 | 1 | $K_0$ |
| c | $K_0$ | 0 | $X_t/X_{max}$ | $K_0$ |
| d | $K_{max}$ | $\gamma \cdot K_{max}$ | $(X_t/X_{max})^{BE}$ | $K_{max}$, $\gamma$, BE |

Table 6: Streamflow error statistics for calibration period, evaluation period, and total simulation period (in parenthesis) for all four different cases in Stage II-1 analysis.

| Calibration | | | | |
|---|---|---|---|---|
| **Metrics** | **Step II-1a** | **Step II-1b** | **Step II-1c** | **Step II-1d** |
| **NMSE** | 0.56 | 0.52 | 0.51 | 0.51 |
| **NBμ** | 0.09 | 0.10 | 0.10 | 0.10 |
| **NBσ** | -0.12 | -0.19 | -0.04 | -0.04 |
| **R** | 0.76 | 0.77 | 0.80 | 0.80 |
| Evaluation (Total Simulation) | | | | |
| **Metrics** | **Step II-1a** | **Step II-1b** | **Step II-1c** | **Step II-1d** |
| **NMSE** | 0.84 (0.77) | 0.65 (0.77) | 0.88 (0.72) | 0.84 (0.70) |
| **NBμ** | 0.14 (0.14) | 0.07 (0.16) | 0.14 (0.14) | 0.14 (0.13) |
| **NBσ** | -0.04 (-0.08) | 0.00 (-0.06) | 0.16 (0.10) | 0.16 (0.10) |
| **R** | 0.66 (0.72) | 0.70 (0.75) | 0.73 (0.78) | 0.74 (0.78) |



Table 7: Streamflow and AET error statistics in total simulation for all four cases in Step II-1.

| Metric | Step II-1a | Step II-1b | Step II-1c | Step II-1d |
|---|---|---|---|---|
| | **Streamflow** | | | |
| **NMSE** | 0.77 | 0.77 | 0.72 | 0.70 |
| **NBμ** | 0.14 | 0.16 | 0.14 | 0.13 |
| **NBσ** | -0.08 | -0.06 | 0.10 | 0.10 |
| **R** | 0.72 | 0.75 | 0.78 | 0.78 |
| | **AET** | | | |
| **NMSE** | 8.93 | 1.28 | 1.70 | 1.71 |
| **NBμ** | 0.18 | 0.12 | 0.17 | 0.17 |
| **NBσ** | 1.89 | -0.26 | -0.10 | -0.13 |
| **R** | 0.22 | 0.43 | 0.48 | 0.49 |

Table 8: Streamflow error statistics for calibration, evaluation, and total simulation (in parenthesis) for all five different cases in Stage II analysis.

| | **Calibration** | | | | |
|---|---|---|---|---|---|
| **Metrics** | **Step II-1** | **Step II-2** | **Step II-3** | **Step II-4** | **Step II-5** |
| **NMSE** | 0.51 | 1.82 | 0.64 | 0.51 | 0.50 |
| **NBμ** | 0.10 | 0.32 | 0.12 | 0.10 | 0.10 |
| **NBσ** | -0.04 | 0.16 | -0.19 | -0.04 | -0.04 |
| **R** | 0.80 | 0.81 | 0.74 | 0.80 | 0.80 |
| | **Evaluation (Total Simulation)** | | | | |
| **Metrics** | **Step II-1** | **Step II-2** | **Step II-3** | **Step II-4** | **Step II-5** |
| **NMSE** | 0.84 (0.70) | 2.46 (2.29) | 0.92 (1.19) | 0.84 (0.70) | 0.84 (0.70) |
| **NBμ** | 0.14 (0.13) | 0.37 (0.36) | 0.10 (0.22) | 0.14 (0.14) | 0.14 (0.13) |
| **NBσ** | 0.16 (0.10) | 0.38 (0.31) | 0.05 (-0.01) | 0.16 (0.10) | 0.16 (0.09) |
| **R** | 0.74 (0.78) | 0.70 (0.77) | 0.61 (0.69) | 0.74 (0.78) | 0.74 (0.78) |





**FIGURES**

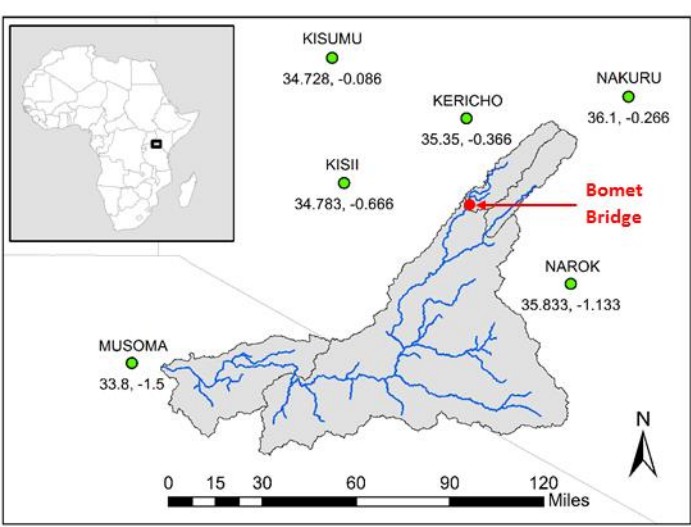

**Fig. 1.** The Mara River basin and the Nyangores River sub-basin. The discharge station is located at Bomet Bridge (red dot).

Meteorological stations (green dots) are located in the surrounding regions (Adapted from Roy et al., *in prep*).

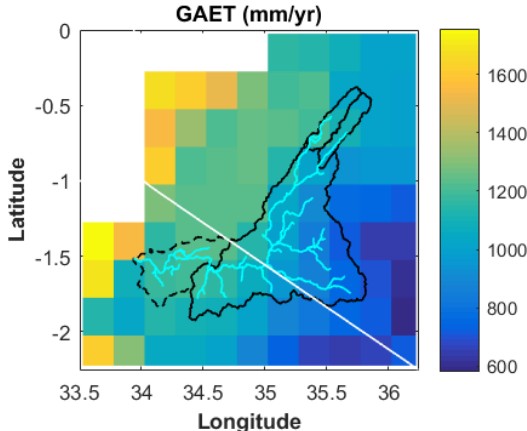

**Fig. 2.** Annual mean of GAET over the entire MRB.





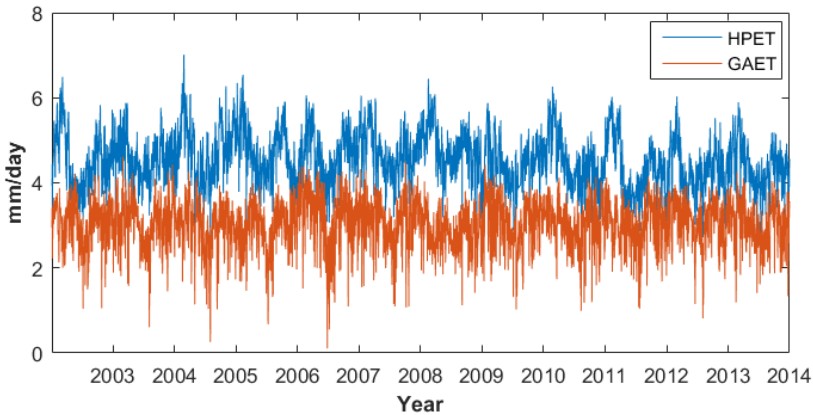

**Fig. 3.** Time series of HPET and GAET for the NRB.

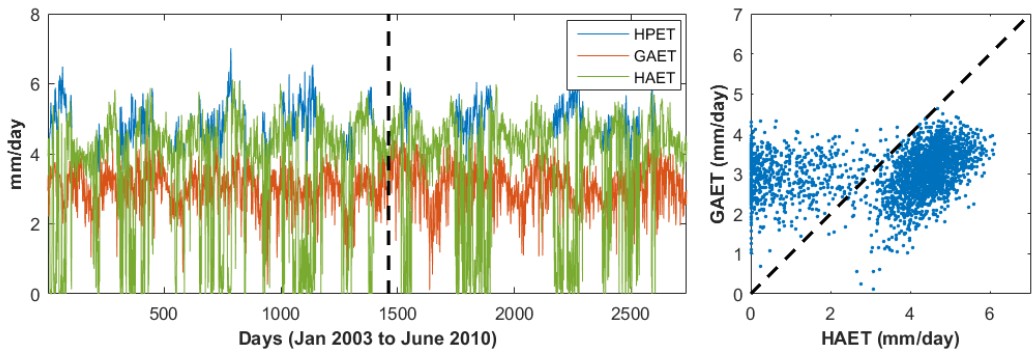

**Fig. 4.** Time series and scatter plots of HPET, GAET, and HAET.

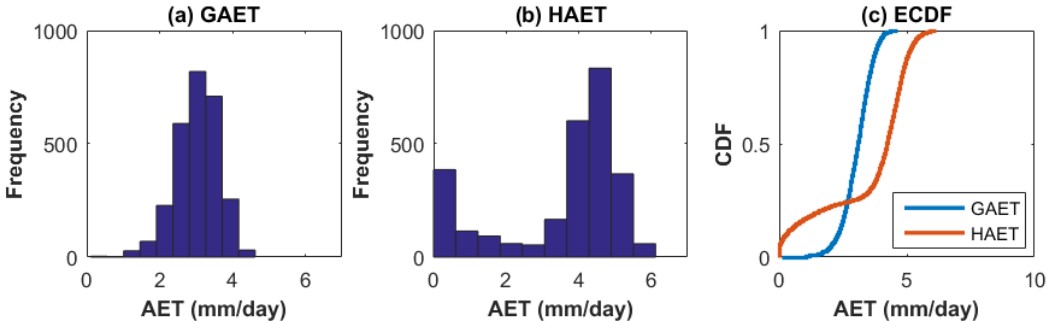

**Fig. 5.** Histogram and ECDF plots of GAET and HAET.



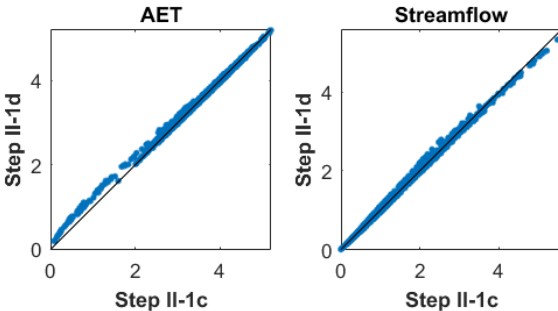

**Fig. 6.** Scatter plots of streamflow and AET from Step II-1a and Step II-1b.

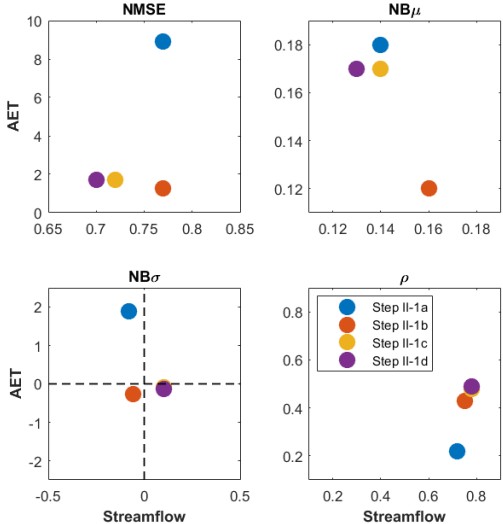

**Fig. 7.** Streamflow and AET error statistics in total simulation for all four cases in Step II-1.





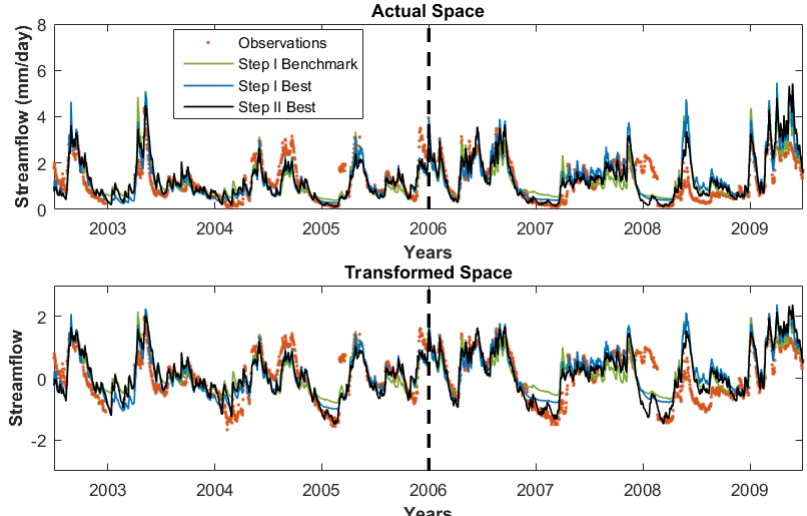

**Fig. 8.** Time series plots of streamflow for the best simulations in Step I and Step II, the benchmark simulation, and the observations.

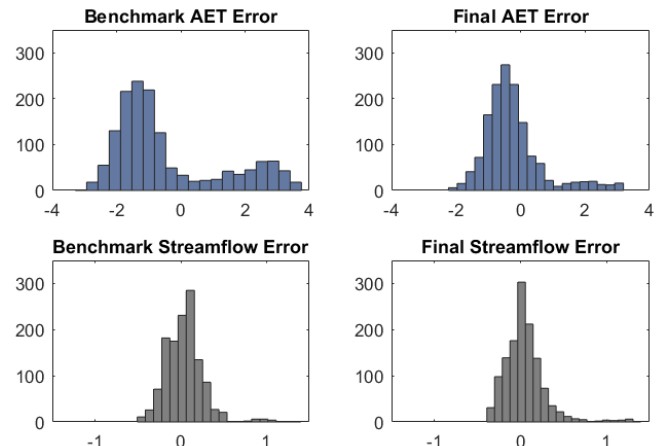

**Fig. 9.** AET and streamflow error distributions for the benchmark and the final steps.

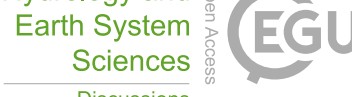

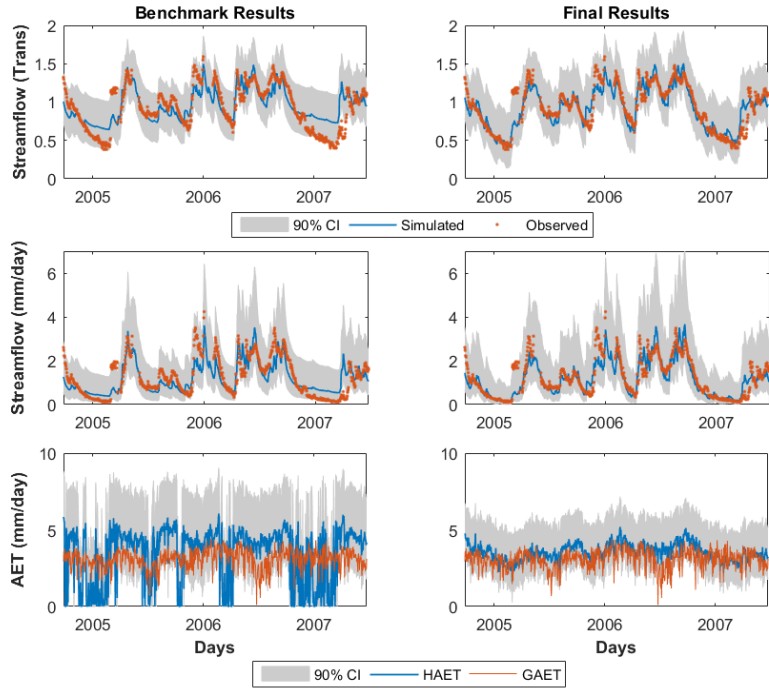

**Fig. 10.** Time series plots of streamflow (λ-transformed and actual) and AET for the benchmark and the final (Stage II-1d) steps.
For clarity, we only show a window of 1000 days.