# Peer review of "Using Satellite-Based Evapotranspiration Estimates to Improve the Structure of a Simple Conceptual Rainfall-Runoff Model"

_Hydrology and Earth System Sciences, 2016_

## Referee Comment (RC1) · A. Bahremand (Referee) · 12 Sep 2016

Referee comments:

1. General comments:

- Comments for the authors:

I congratulate you on this work. You have nicely shown that with the help of new information we can improve our hydrological models (as you called it the "diagnostic structural improvement") in a physically meaningful manner to be able to apply them for ungauged basins with higher confidence and more certainty. I like your paper (especially for its attempt to get good results by intellectually satisfying modeling [and with

"limited calibration"]), I only wish you could add one or two sentences into the paper abstract of what you have written in your discussion and overall outlook (I know it may have word number limitation). There are some little corrections which I have mentioned them on the attached pdf file. Please note that the pdf should contain 9 comments, 5 direct corrections written on the text, and 7 highlighted or underlined words or phrases (if not please email me to list them separately for you).

- Comments for the editor:

In my opinion, the paper deserves publication with some minor corrections. It is a good research in line with the recent works. It has novelty and promotes "meaningful modeling". It uses new satellite based information to improve a hydrologic model and to show how new information can reduce uncertainty, and can direct the parameter estimation in a physically meaningful manner, and how we can improve our hydrological model structurally enabling the application of our models for ungauged basins with more confidence. The paper is very well written (in terms of structure and English both). I agree with all results, figures, tables, and the arguments around the results, and the valuable discussion afterwards. I thank you for giving me the opportunity to review this good paper.

2. Specific comments:

If possible adding one concluding sentence into your abstract (similar of what you have written in your discussion, overall outlook and conclusion). some corrections are listed on the attached pdf. Please note that the pdf should contain 9 comments, 5 corrections written directly on the text, and 7 highlighted or underlined words or phrases.

At the end, again, I thank the authors and the editor Prof. Saco, for their work on this paper and letting me review it. (I may further attend in the posssible future interactive discussion related to the paper)

Best regards,

Abdolreza Bahremand

Please also note the supplement to this comment:
http://www.hydrol-earth-syst-sci-discuss.net/hess-2016-413/hess-2016-413-RC1-supplement.pdf

---

## Referee Comment (RC2) · A. Bahremand (Referee) · 15 Sep 2016

Dear colleagues,

In my previous post, the sticky notes of the attached file were not readable. Again, I wrote each comment directly on the attached pdf file (with purple color). However, to make it sure this time I also listed all those comments (sticky notes) here below:

The sticky notes:

1. The paper and its discussion and conclusion has much more other useful contents than what has been given in the abstract. I guess the limitation of the abstract word numbers (500 words) has been the reason for this.

[Figure]

2. One concluding sentence (like those written in the conclusion) should be added here.

3. GLEAM and HyMod could be other keywords for this paper? Don't you think so?

4. Here, in such case I am sure the authors know better than me that the strong correlation is not enough :)

5. , and whether the model provides improved ..........[here, i proposed this little correction to make it in harmony with the previous sentences. So, i think adding the conjunction "whether", like what you did for the previous sentence, is better here.]

6. Unnecessary abbreviation makes the text a bit boring, as the text has already too many :) If you wish to reduce them then just start with the name of the rivers...

7. The abbreviation, HAET, has not been introduced in the paper so far. Perhaps you mean HyMod AET. First I thought that H stands for Hargreaves...

8. after the remove of the GAET data.

9. This sentence needs a little bit of improvement (grammar correction).

I am sorry for the inconvenience,

Best Regards,

Abdolreza Bahremand

Please also note the supplement to this comment:
http://www.hydrol-earth-syst-sci-discuss.net/hess-2016-413/hess-2016-413-RC2-supplement.pdf

[Figure]

**Supplement:**

Hydrology and Earth System Sciences

**Using Satellite-Based Evapotranspiration Estimates to Improve the Structure of a Simple Conceptual Rainfall-Runoff Model**

Tirthankar Roy1, Hoshin V. Gupta1, Aleix Serrat-Capdevila1,2 and Juan B. Valdes1

[revised manuscript text omitted]

even when GLEAM-satellite-based evapotranspiration estimates are no longer available. There, above, i proposed this little correction to make it in harmony with the previous sentences. So, i think adding the conjunction "whether", like what you did for the previous sentence, is better here. 2 Study Area, Data and Methodology

**2.1 Study Area**

**basin/**

- 5 [8] This study is carried out for the Nyangores River Basin (NRB), which is a sub-basin of the Mara River flowing through Kenya (Fig. 1). NRB has an aerial coverage of 697 km2 and is located at the northeastern side of the Mara River Basin (MRB; Location: 33°88'E 35°90'E 0°28'S 1°97'S). The perennial Nyangores River originates from the Mau Escarpment (3000 m ASL) fault scarp passing through the western side of the Great Rift Valley in Kenya. It then merges with the Amala River at the Napuiyapi swamp (2932 m ASL) to form the Mara River, which flows all the way to Lake Victoria at Musoma Bay, Tanzania
- 10 (1130 m ASL). MRB (or NRB) has two wet seasons consequent to the yearly oscillations of the inter-tropical convergence zone (ITCZ), the primary wet season occurring during March to May (MAM) and the secondary during October to December (ONL). The long-term mean rainfall in the Mau Escarpment is around 1500 mm. The rainfall in the basin is influenced by factors like

topography, elevation gradient, regional influence of Lake Victoria, sea-surface temperature (SST) of the Indian Ocean, etc. Unnecessary abbreviation makes the text a bit boring, as the text has already too many :) If you wish to reduce them then just start with the name of the rivers in the rivers in the river in t

[revised manuscript text omitted]
- Step I-3:
   Recalibrated GAET Constrained: GAET is used to constrain HyMod-V1 HAET. The model parameters are recalibrated to match simulated to observed streamflows.
  - Step I-4: Bias-Corrected Recalibrated GAET Constrained: A "bias-correction" is applied to GAET, and the biascorrection and model parameters are calibrated together to match simulated and observed streamflows. Since no ground-based data are available to bias-correct GAET, we tested two empirical bias correction schemes (shown below):

$$y = f(x, \cdots)$$
 and  $x = \boldsymbol{a} \cdot GAET$  (1)

$$y = f(x, \cdots)$$
 and  $x = \boldsymbol{a} \cdot GAET^{\boldsymbol{b}}$  (2)

[revised manuscript text omitted]

---

## Referee Comment (RC3) · Anonymous Referee #2 · 26 Sep 2016

Using Satellite-based evapotranspiration estimates to improve the structure of a simple conceptual rainfall-runoff model

By Roy, Gupta, Serrat-Capdevila, Valdes

This paper presents results from a study examining the use of satellite estimates of actual evapotranspiration (SET) to firstly constrain and secondly modify a HyMod model of Nyangores River Basin in Kenya. Although the ideas presented here are interesting, I found that the reasoning used in the study was circular and I'm not convinced by the results. I think the presentation of the material is too much like a report and the method and results are often mixed up, with the vast majority of the method discussion provided in Section 3 which is nominally the results section. The paper also refers to another

publication in preparation by the same authors on this catchment and without seeing this it is difficult to understand the similarity and any potential overlaps between the two publications. It's not clear why this paper would be presented first. I recommend that the paper is rejected and the authors undertake more extensive validation of the method in a catchment where there is data other than the SET to allow comparisons.

If I understand the method properly, in Case 1 HyMod is run and the AET from the model is found to be different from the SET estimates. So the model is run using SET to constrain the AET in the model by setting the requirement that the AET  $\leq$ SET. However then the model parameters are found to be unrealistic so the SET is bias corrected so that when the model is constrained to have AET

property is not biased which is key for this method and even line 23 where the annual bias is low doesn't guarantee that there are not other biases that are cancelling out throughout the year.

Page 4 – paragraph 12 – TRMM data is no longer available so not clear why you say that it is available to near-present? The study period is not clear from Section 2 in any case.

Page 4, line 34 – here you describe Stage 1 as "constraining" and you are at pains to point out that it is not assimilation and yet in the remainder of the manuscript you continue to use the term assimilation – I think you need to be more careful with the terminology e.g. Page 8, line 23; Page 12, Line 24

Page 5, Step 1-2 – given this is the method section, there are no details here of the actual constraints. These are provided in the results section. I think this makes the presentation quite confused and doesn't provide the reader with much of a sign post or guide as where the research is heading. Similar comments for Step II-1 where the four equations are mentioned.

Page 7, Line 24 – I don't understand why you validate your water balance using satellite precipitation which has its own concerns. Why not use some ground based data as well?

Page 7, Line 27 – "based on our expectation of how it would behave" – this comes to my concern about the validation. We generally expect a more robust validation than just a sense that the soil moisture should be smooth. Why should it be smooth for this catchment? You don't appear to have any soil moisture data to validate this statement.

HESSD
**Discussion** paper

---

## Author Comment (AC1) · 5 Oct 2016

We thank the reviewer, who identified himself as Prof. Abdolreza Bahremand, for nicely summarizing the key aspects of our study and pointing out their importance. We have addressed all of his comments and made the suggested changes in our revised manuscript.

NOTE: [1] The manuscript with tracked changes is uploaded in the form of a supplement. [2] Page and line numbers mentioned in the response correspond to the revised manuscript. [3] We have added an additional figure (Fig. 4) to demonstrate the structure of the modified model (HyMod-V2).

————————————————————————————————————————

Reviewer Comment: 1. The paper and its discussion and conclusion has much more other useful contents than what has been given in the abstract. I guess the limitation of the abstract word numbers (500 words) has been the reason for this. 2. One concluding sentence (like those written in the conclusion) should be added here.

Author Response: As per the reviewer's suggestion, we have modified the last sentence of the abstract as in the following to summarize our main outlook: [Page 1 Line 15-17]

"Results suggest that both the approaches can provide improved simulations of streamflow, whereas the second approach also significantly improves the simulations of actual evapotranspiration, which substantiates the importance of 'diagnostic structural improvement' of hydrologic models."

————————————————————————————————————————

Reviewer Comment: 3. GLEAM and HyMod could be other keywords for this paper? Don't you think so?

Author Response: We agree with the reviewer on this and have now included both GLEAM and HyMod as keywords.

————————————————————————————————————————

Reviewer Comment: 4. Here, in such case I am sure the authors know better than me that the strong correlation is not enough :)

Author Response: Yes, we agree that a strong correlation is not enough. Therefore, in the revised manuscript, we have now included a detailed discussion on the evaluation of GLEAM. We are now citing one book chapter and four papers for this discussion. [Page 2 Line 21 – Page 3 Line 4]

". . . Worldwide evaluations suggest that satellite-based ET estimates are strongly correlated (∼0.83) with ground-based observations made at flux towers (Demaria and Serrat-Capdevila, 2016). We use the Global Land Evaporation Amsterdam Model (GLEAM) as the source of the satellite-based ET (SET) data for this study. In GLEAM algorithm, ET is computed using only a small number of satellite-based inputs, which is largely beneficial for sparsely gauged basins. Miralles et al. (2011) have shown that GLEAM estimates of evaporation are strongly correlated (0.80) with annual cumulative evaporation estimated via eddy covariance at 43 stations, and have very low (-5%) average bias. The correlations at individual stations are strong (0.83) for all vegetation and climate conditions, and improve to 0.9 for monthly time series (Miralles et al., 2011). McCabe et al. (2016) have found satisfactory statistical performance (R2 = 0.68; Root Mean Square Difference = 64 Wm−2; Nash-Sutcliffe Efficiency = 0.62) of GLEAM while compared against the data from 45 globally-distributed eddy-covariance stations. Michel et al. (2016) compared Priestley-Taylor Jet Propulsion Laboratory model (PT-JPL), Moderate Resolution Imaging Spectroradiometer evaporation product (PM-MOD), Surface Energy Balance System (SEBS), and GLEAM simulations against 22 FLUXNET tower-based flux observations and found GLEAM and PT-JPL to be more closely matching the in-situ observations for the selection of towers and the reference period (2005-2007). Their extended analysis with 85 towers had similar overall outcomes. Miralles et al. (2016) compared three process-based ET methods (PM-MOD, GLEAM and PT-JPL) against surface water balance from 837 globally distributed catchments, and reported that GLEAM and PT-JPL provide more realistic estimates of ET. They found these two products to provide superior overall performance for most ecosystem and climate regimes, while PM-MOD tends to underestimate the flux in tropics and subtropics."
* * *
Reviewer Comment: 5. , and whether the model provides improved ..........[here, i proposed this little correction to make it in harmony with the previous sentences. So, i think adding the conjunction "whether", like what you did for the previous sentence, is

better here.]

Author Response: Thanks for the suggestion. We have now modified the sentence as: [Page 3 Line 16-17]

"Finally, we test whether the use of GLEAM SET can further improve the performance of the structurally modified model, and whether there is any drop in the performance of the model if GLEAM SET data become unavailable."
* * *
Reviewer Comment: 6. Unnecessary abbreviation makes the text a bit boring, as the text has already too many :) If you wish to reduce them then just start with the name of the rivers...

Author Response: We agree with the reviewer on this. We have now removed the abbreviations for the seasons as well as for the basins.
* * *
Reviewer Comment: 7. The abbreviation, HAET, has not been introduced in the paper so far. Perhaps you mean HyMod AET. First I thought that H stands for Hargreaves...

Author Response: Thanks for pointing this out. We have now introduced HAET in Section 2.3.
* * *
Reviewer Comment: 8. after the remove of the GAET data.

Author Response: We have now modified this part of the sentence as: ". . . after the removal of the GAET data."
* * *
Reviewer Comment: 9. This sentence needs a little bit of improvement (grammar correction).

[Figure]

Author Response: Thanks for pointing this out. We changed the sentence as in the following:

"Therefore, our results suggest that ET constraining approach should be implemented only for the simulation periods when SET data are available." However, this statement seems redundant once the constraining results are already discussed. Therefore, we deleted this from the revised manuscript.

Please also note the supplement to this comment: http://www.hydrol-earth-syst-sci-discuss.net/hess-2016-413/hess-2016-413-AC1-supplement.pdf

―――――――――――――――――

---

## Author Comment (AC2) · 5 Oct 2016

We thank the reviewer for reviewing our manuscript and providing his/her valuable feedbacks. We have now addressed all of his/her comments and discussed them in the following. As the reviewer mentioned, there were some places in the manuscript which created confusions and the concepts seemed circular. We agree with the reviewer on that. These were mainly due to the lack of sufficient care in the use of terminology. We have revised the manuscript to resolve these issues and make our message more clear-cut. Thanks to the reviewer's feedback, the paper is now much improved.

NOTE: [1] The manuscript with tracked changes is uploaded in the form of a supplement. [2] Page and line numbers mentioned in the response correspond to the revised

manuscript. [3] We have added an additional figure (Fig. 4) to demonstrate the structure of the model.
* * *
Reviewer Comment 1: This paper presents results from a study examining the use of satellite estimates of actual evapotranspiration (SET) to firstly constrain and secondly modify a HyMod model of Nyangores River Basin in Kenya. Although the ideas presented here are interesting, I found that the reasoning used in the study was circular and I'm not convinced by the results. I think the presentation of the material is too much like a report and the method and results are often mixed up, with the vast majority of the method discussion provided in Section 3 which is nominally the results section. The paper also refers to another publication in preparation by the same authors on this catchment and without seeing this it is difficult to understand the similarity and any potential overlaps between the two publications. It's not clear why this paper would be presented first. I recommend that the paper is rejected and the authors undertake more extensive validation of the method in a catchment where there is data other than the SET to allow comparisons.

Author Response 1: The manuscript is designed such that all the analyses steps are clearly stated and their results are thoroughly discussed. This is important since we recommend this approach for similar investigations, due to the fact that it's inclusive. It takes into account several important issues, including process constraining, use of constraint adjustment, usefulness of model (re)calibration, information assimilation (from satellite-based sources), diagnostic model structural improvement, and uncertainty analysis. However, as pointed out by the reviewer, we do see that some method discussions could be removed from the results section and put back to the methods section itself. We have now taken care of this issue in our revised manuscript.

Regarding the point on the second publication, we do have another manuscript under review, however, we would like to clarify that the objective and scope of that manuscript

are quite different as compared to this one. That manuscript reports on the development of a multi-model and multi-product (satellite)-based probabilistic operational streamflow forecasting platform for sparsely-gauged basins and does not in any way address the problem of model structural correction/improvement. We are ready to share the manuscript with the reviewer and editor personally if necessary to resolve this concern.

Since the other manuscript is under review, we are not citing that anymore in this manuscript.

Regarding the comment on the other available data for comparison, note that the dataset (GLEAM) we are using has already been validated in several recent studies. Although we didn't include the detailed discussion on validation in our initial manuscript, we have now included that part in our revised manuscript (Page 2 Line 21 - Page 3 Line 4). GLEAM has already been evaluated both at local (eddy-covariance towers) and global scales. There have been projects that have focused on the topic of the evaluation of GLEAM, e.g. The WAter Cycle Multi-mission Observation Strategy-EvapoTranspiration (WACMOS-ET), Global Energy and Water Cycle Exchanges (GEWEX) LandFlux Project, etc.

All the studies cited in our revised manuscript (one book chapter and four paper) found GLEAM to be one of the best ET products. Therefore, we don't think it is necessary to carry out an additional evaluation of GLEAM, given the fact that other studies have already focused on that part. This also does not fit well with the main goals of this manuscript. Moreover, an evaluation study of this kind would stand out on its own as an independent paper, which is clearly beyond the scope of this manuscript.

To be clear, the main objective of this study is NOT to validate/compare actual ET products, which is an interesting topic, but appropriate for a different manuscript. In this study, we explore different structure-related methods (including process constraining) to improve the performance of a rainfall-runoff model, and we have an inclusive design

to organize all the steps in a systematic manner. We show how the model deficiencies could be overcome by using new sources of information.
* * *
Reviewer Comment 2: If I understand the method properly, in Case 1 HyMod is run and the AET from the model is found to be different from the SET estimates. So the model is run using SET to constrain the AET in the model by setting the requirement that the AET <= SET. However then the model parameters are found to be unrealistic so the SET is bias corrected so that when the model is constrained to have AET <= SET, the model parameters are more realistic. In all of this there is no evaluation of the SET itself and the bias correction step implies that there are problems with the SET. So you're trying to match a model to a biased quantity and then changing that quantity and then still trying to match it. It just seems very circular to me. Case 2 follows much the same logic except rather that using the constraint that AET <= SET, the model structure is changed with a variety of different equations that factor the evaporative demand ratio. Finally in Figure 9 the model is compared back to the SET which was used to correct the model I just don't understand how you can accept the SET data without having an external validation. I accept that this is unlikely to exist for the catchment you have chosen but I think you then need to test your method in a more instrumented catchment where you do have external validation data and once you have confidence in the method then you can apply to a poorly gauged basin.

Author Response 2: This is a very interesting point which we unfortunately did not explain well in the original manuscript. We should point out that there is no 'bias correction' in this study. For a proper correction, we need the 'ground truth' which we don't have in our case for ET. Therefore, the term 'bias-correction' was wrongly used and we have now changed that. We are now calling it 'constraint adjustment' because that is what it is actually doing. In Stage 1, the model structure is fixed. When GAET is used as a constraint in the ET process within the model, it introduces bias in the stream-flow. Therefore, we adjust the constraint such that that bias is removed. Note that this

is NOT indicative of the presence of any actual bias within the GAET estimates. The constraint adjustment factor is a model "parameter" which corresponds to the structural deficiencies of the model. It may or may not be necessary as the structure changes. In Stage 2, we saw that when the structure was improved (deficiencies reduced), ET constraining became irrelevant.

Regarding the point of external validation, please see the last three paragraphs of our first response.

————————————————————————————————————————

Reviewer Comment 3: Page 2 – paragraph 3 – at this stage its not clear how ET can be a model target – I think you need to make it clearer at this point that PET is forcing data and AET is a model state.

Author Response 3: We consider precipitation and PET as the forcings. Note that the precipitation is the only input to the water budget of the model, PET is a constraint to set the upper limit of the actual ET. The model produces both discharge and actual ET as outputs. Therefore, we don't see why AET needs to be considered as a model "state" (as conventionally defined). It is clearly a model simulated "output".

————————————————————————————————————————

Reviewer Comment 4: Page 2, line 15 – good correlation of the SET does not give me confidence that the property is not biased which is key for this method and even line 23 where the annual bias is low doesn't guarantee that there are not other biases that are cancelling out throughout the year.

Author Response 4: We agree we had a very brief discussion on the comparison/validation of the ET products in our initial manuscript. We have now expanded that discussion in our revised manuscript, where some additional error statistics (apart from correlation coefficient) are also reported.

Page 2 Line 21 – Page 3 Line 4 "... Worldwide evaluations suggest that satellitebased ET estimates are strongly correlated (∼0.83) with ground-based observations made at flux towers (Demaria and Serrat-Capdevila, 2016). We use the Global Land Evaporation Amsterdam Model (GLEAM) as the source of the satellite-based ET (SET) data for this study. In GLEAM algorithm, ET is computed using only a small number of satellite-based inputs, which is largely beneficial for sparsely gauged basins. Miralles et al. (2011) have shown that GLEAM estimates of evaporation are strongly correlated (0.80) with annual cumulative evaporation estimated via eddy covariance at 43 stations, and have very low (-5%) average bias. The correlations at individual stations are strong (0.83) for all vegetation and climate conditions, and improve to 0.9 for monthly time series (Miralles et al., 2011). McCabe et al. (2016) have found satisfactory statistical performance (R2 = 0.68; Root Mean Square Difference = 64 Wm−2; Nash-Sutcliffe Efficiency = 0.62) of GLEAM while compared against the data from 45 globally-distributed eddy-covariance stations. Michel et al. (2016) compared Priestley-Taylor Jet Propulsion Laboratory model (PT-JPL), Moderate Resolution Imaging Spectroradiometer evaporation product (PM-MOD), Surface Energy Balance System (SEBS), and GLEAM simulations against 22 FLUXNET tower-based flux observations and found GLEAM and PT-JPL to be more closely matching the in-situ observations for the selection of towers and the reference period (2005-2007). Their extended analysis with 85 towers had similar overall outcomes. Miralles et al. (2016) compared three process-based ET methods (PM-MOD, GLEAM and PT-JPL) against surface water balance from 837 globally distributed catchments, and reported that GLEAM and PT-JPL provide more realistic estimates of ET. They found these two products to provide superior overall performance for most ecosystem and climate regimes, while PM-MOD tends to underestimate the flux in tropics and subtropics."
* * *
Reviewer Comment 5: Page 4 – paragraph 12 – TRMM data is no longer available so not clear why you say that it is available to near-present? The study period is not clear from Section 2 in any case.

Author Response 5: This is a good point. We have now included this information into our revised manuscript. We are using the TRMM Multi-Satellite Precipitation Analysis (TMPA-RT) dataset which is still available. This is a merged dataset. TRMM Microwave Imager (TMI) was a part of it, which is no more operational (since 8 April 2015) because of fuel and battery issues with the satellite. As mentioned by the developers, the absence of TRMM is not crucial to the production of TMPA and TMPA-RT data.

We discuss the time periods in Section 2.5:

"The model was run continuously for the 7.5-year period Jan 2003 to June 2010, with the first 4 years (2003 to 2006) used for calibration and the remaining 3.5 years (2007 to mid-2010) used to provide an additional assessment of model performance. Results are shown for the "calibration (4-years)", "evaluation (3.5 years)" and "total (7.5 years)" simulation periods."
* * *
Reviewer Comment 6: Page 4, line 34 – here you describe Stage 1 as "constraining" and you are at pains to point out that it is not assimilation and yet in the remainder of the manuscript you continue to use the term assimilation – I think you need to be more careful with the terminology e.g. Page 8, line 23; Page 12, Line 24

Author Response 6: Thank you for pointing this out. We have now removed the term 'assimilation' wherever required.
* * *
Reviewer Comment 7: Page 5, Step 1-2 – given this is the method section, there are no details here of the actual constraints. These are provided in the results section. I think this makes the presentation quite confused and doesn't provide the reader with much of a sign post or guide as where the research is heading. Similar comments for Step II-1 where the four equations are mentioned.

Author Response 7: We agree with the reviewer on this and have now restructured the
methods and results sections in our revised manuscript.
* * *
Reviewer Comment 8: Page 7, Line 24 – I don't understand why you validate your water balance using satellite precipitation which has its own concerns. Why not use some ground based data as well?

Author Response 8: Note that the TMPA data used in this study has been bias corrected using rain gauge measurements from the study area. The detailed methodology is discussed in the other paper. As mentioned earlier, we are ready to share the other manuscript personally with the reviewer or the editor.
* * *
Reviewer Comment 9: Page 7, Line 27 – "based on our expectation of how it would behave" – this comes to my concern about the validation. We generally expect a more robust validation than just a sense that the soil moisture should be smooth. Why should it be smooth for this catchment? You don't appear to have any soil moisture data to validate this statement.

Author Response 9: Thanks for pointing this out. We agree that in order to make this statement, the soil moisture data need to be studied first. Therefore, we have now removed this sentence from our revised manuscript.
* * *
Please also note the supplement to this comment:
http://www.hydrol-earth-syst-sci-discuss.net/hess-2016-413/hess-2016-413-AC2-supplement.pdf

**Supplement:**

[revised manuscript text omitted]

have found satisfactory statistical performance ($R^2$ = 0.68; Root Mean Square Difference = 64 $Wm^{-2}$; Nash-Sutcliffe Efficiency = 0.62) of GLEAM while compared against the data from 45 globally-distributed eddy-covariance stations. Michel et al. (2016) compared Priestley-Taylor Jet Propulsion Laboratory model (PT-JPL), Moderate Resolution Imaging Spectroradiometer evaporation product (PM-MOD), Surface Energy Balance System (SEBS), and GLEAM simulations against 22 FLUXNET tower-based flux observations and found GLEAM and PT-JPL to be more closely matching the in-situ observations for the selection of towers and the reference period (2005-2007). Their extended analysis with 85 towers had similar overall outcomes. 
[revised manuscript text omitted]

**4 Discussion and Conclusions**

5      This study has explored two different approaches to the use of recently available SET data from GLEAM to improve the realism and performance of the conceptual catchment-scale hydrologic model HyMod. In the first approach, SET data were used to constrain the ET estimates, while in the second we modified the model structure itself. Our study shows that use of satellite-based information can clearly benefit the process of hydrologic modeling for poorly gauged basins by providing new sources of information to reduce the epistemic component of model structural uncertainty through improved physical process representation.

**4.1 Constraining ET**

The Use of ET data as a to constraint  can improve streamflow forecasts, provided some additional processing steps are implemented. Direct insertion of GAET into the ET equation of the HyMod model resulted in bias (Step I-2); the type of bias will, of course, be subject to change depending on the SET data used. While recalibration of the model improved the  performance (Step I-3), it resulted in a conceptually unrealistic estimate for the storage capacity of the basin. Application of constraint adjustments on GAET  improved the streamflow forecasts and also resulted in a more realistic estimate for basin storage capacity (Step I-4); use of multiplicative and power-law type adjustment  schemes produced almost similar results, with slightly better error statistics for the latter. The streamflow simulations of the model from Step I-4 do not deteriorate if the GAET data become unavailable.

[revised manuscript text omitted]

---

## Author Comment (AC4) · 13 Oct 2016

Author Comments: General

Our study explores two (three after combining) different ways of incorporating satellite-based actual ET (AET) data into conceptual hydrologic modeling framework to improve its performance in terms of streamflow and AET simulations:

(i) Based on constraining the ET process within the model. (ii) Based on changing the model structure to improve the ET process parameterization. (iii) Combination of both (constraining and structural improvement).

Following are some key points that we think are unique about this study:

[1] We have followed a strategy that includes several important aspects of modeling, such as: (a) Process constraining (including the new constraint adjustments) (b) Model re-calibration (using SCE-UA) (c) Use of new sources of information (d) Diagnostic model structural improvement and (e) Uncertainty analysis

[2] Diagnostic structural improvement has now become a new focus in the field of hydrology, where earlier the main focus was only on model calibration. A recent paper on this topic is Gupta et al. (2012). In this study, our main focus it to correct the model structure first and then calibrate the model parameters. We have tested several structural forms of the ET process parameterization equation within the model with increasing complexity and selected the one that performed the best.

[3] Although potential ET (PET) data are frequently available, the availability of actual ET (AET) data has always been very limited. The AET product we used (GLEAM) is quite new. We are using the latest version of GLEAM satellite-only AET which was made available few months ago.

[4] This study shows how the new sources of information (satellite AET in this case) can be utilized within the hydrologic modeling framework to improve its performance. The model HyMod has been there for a long time, however, only after the availability of the new AET datasets, the model could be improved.

[5] We think that both structural improvement and model calibration are equally important for models that have practical applications, and in this study, we apply both. We change the structure of the ET process parameterization equation and then calibrate the model against streamflow. Note that we deliberately did not calibrate the model against AET. The main reason for that was to show that if the model structure is changed, the importance of calibration is significantly reduced. Prof. Abdolreza Bahremand, who is also the first reviewer of this paper (see his positive comments in the discussion forum), has a recent opinion paper on this topic (Bahremand, 2016).

[6] The final plot of the paper (Figure 11 in the revised manuscript which is also available in the discussion forum) clearly shows the improvements after carrying out the structural modifications within the model. The left column is the original benchmark model and the right column is the final model with structural modifications. As can be seen, the streamflow simulations are more accurate when the model structure is changed. The recession are more accurately simulated. For AET simulations, the improvement is even more prominent. The blue line is the HyMod generated AET. As can be seen, after the structural modification, the AET simulations are way much better. The blue line in the second-column last-row plot is following the red line (GLEAM) more closely and the confidence bounds (grey) are also narrowed down significantly. These results clearly signify the importance of this study.

References

Bahremand, A., 2016. HESS Opinions: Advocating process modeling and de-emphasizing parameter estimation. Hydrol. Earth Syst. Sci. 20, 1433–1445. doi:10.5194/hess-20-1433-2016

Gupta, H. V., Clark, M.P., Vrugt, J.A., Abramowitz, G., Ye, M., 2012. Towards a comprehensive assessment of model structural adequacy. Water Resour. Res. 48, W08301. doi:10.1029/2011WR011044

---

## Short Comment (SC1) · 14 Oct 2016

In the submitted manuscript, results about three different ways of incorporating satellite-based actual ET (AET) data into conceptual hydrologic modeling framework to improve its performance in terms of streamflow and AET simulations are reported (process constraining within the model, structural change, and a combination of both). The key points that I have found are:

This study shows how the new sources of information such as satellite AET can be used to improve the performance of hydrologic models.

A strategy that includes several important aspects of modeling is described, which includes process constraining, model recalibration, use of newly available information, and finally a diagnostic structural improvement in model together with uncertainty analysis.

From the results exposed, both structural improvement and process constraining are important for models that have practical applications This idea would be helpful for further developments in hydrological modeling. From my point of view, this paper represents important novelty in the field of hydrological research and is quite suitable for HESS.

---

## Short Comment (SC2) · 14 Oct 2016

I really appreciate the authors' work that does an excellent job of improving the model performance in terms of two important hydrological processes, ET and streamflow, by using both process constraining and structure retrofit.

This paper begins with the important observation that satellite-based ET (SET) estimates have the potential to be useful for hydrologic applications. Using a new satellite-based AET product, authors constrained the ET, one of the outputs of the model to improve the model performance in a physically consistent process modification.

Then the authors intellectively modified the model structure rather than just recalibrating the model parameters. This nice work shows an authoritative manner of 'diagnostic structural improvement' for a model. Gupta et al. (2012) proposed 'model structure adequacy approach' comprehensively and this work provided a case study to show the ideas in detail.

References

Gupta, H. V., Clark, M.P., Vrugt, J.A., Abramowitz, G., Ye, M., 2012. Towards a comprehensive assessment of model structural adequacy. Water Resour. Res. 48, W08301. doi:10.1029/2011WR011044

---

## Author Comment (AC5) · 14 Oct 2016

We thank Prof. Durán-Barroso for highlighting the key aspects of our study and acknowledging its merits.

---

## Author Comment (AC6) · 14 Oct 2016

Many thanks to Tao Liu for highlighting the key points and indicating the importance of diagnostic structural improvement of a model.

---

## Short Comment (SC3) · 16 Oct 2016

General comments: Evapotranspiration is a very important variable in rainfall-runoff models. To a great extent, evapotranspiration determines the calculation of water balance. Oftentimes the conceptual hydrologic models simulate unrealistic values of evapotranspiration. In this paper, the authors consider different ways of using a newly available evapotranspiration dataset to improve the performance of a conceptual model both in terms of streamflow and evapotranspiration simulations. These avenues are studiedcomprehensively, involving a lot of research and analysis work. The performance of evapotranspiration data is different for different temporal scales as pointed out by the authors in the literature review. The literature review in the

revised manuscript also nicely summarizes the performance of different satellite-based evapotranspiration products. The new satellite-based evapotranspiration information is used to improve a hydrologic model and to show how new information can reduce uncertainty and improve our hydrological model structurally enabling the application of hydrological models for ungauged basins with more confidence. The paper is very well written in both structure and English. However, at the first read of this paper, I had two followed questions about this article: (1) The authors show that the soil moisture should be smooth in the research basin. Why should it be smooth for this catchment? (2) I don't understand why you validate your water balance using satellite precipitation which has its own concerns. But after I read all of the review comments, response of all comments and the revised manuscript carefully, I found that the authors have made perfect response to the previous two reviewers' comments, and answered these two above mentioned questions very well. View from my angle, the research results of this paper can increase the amount of information for the rainfall-runoff model when it was applied in poorly gauged basins. So, the paper deserves publication in this journal. It's a good research work which has innovation, and the results are quite promising.

Please also note the supplement to this comment:
http://www.hydrol-earth-syst-sci-discuss.net/hess-2016-413/hess-2016-413-SC3-supplement.pdf

―――――――――――――――――

---

## Author Comment (AC7) · 17 Oct 2016

We thank Dr. Si for providing his valuable comments. We are also happy to know that we have addressed the questions properly in the review response and the revised manuscript.
* * *

---

## Short Comment (SC4) · 18 Oct 2016

This article explores three approaches to use newly available actual ET (AET) data from satellite products into conceptual hydrologic models to improve predictions of streamflow and AET of simple hydrological models:

1. Calibrate the model with AET data 2. Change model structure to improve ET process parameterization 3. Combination of both

The authors compare the performance of the three approaches and concluded by stressing the importance of structural medications within the model. By only calibrating the model with AET data, the improvements from initial estimates of the state does not

sustain into future time steps. But if the GAET data is used to modify model structure, such improvements prevail over subsequent time steps. As the authors have rightly pointed out in the article, inadequacy of model structure tend cause the state estimates to drift away from their ideal ("ideal" is a confusing choice of word, pp13) values over time so that model performance deteriorate with increasing time away from the model calibration period. I think the above is consistent with the parsimony principle (i.e. Occam razor). With new information available (in this case AET data), more complex models may become more necessary. The take-home message from this exercise to all modellers is that when presented with new data, they should double check whether the current model is too simple. Strategies that allow increasing model complexity like the one in the present study are needed.

In my opinion, I think this work make important contributions in terms of using a new kind of data (satellite-based AET), as well as how to best use it (structural modifications). Its findings will shed light on future efforts to bring in more remote sensing data for regional hydrological predictions. Specific comments:

1. I believe there may be inconsistency in symbols for correlation coefficient. In Table 1, you use rho for correlation coefficient, but R in subsequent tables.

---

## Author Comment (AC8) · 18 Oct 2016

We thank Mr. Tso for his valuable comments on our manuscript and nicely pointing out what we should do when we have new sources of information available.

Regarding his specific comments:

[1] We agree that the word 'ideal' is probably not the best choice here. We are now using the word 'appropriate' instead of 'ideal'.

[2] We are now using consistent symbol for the correlation coefficient.

---

## Referee Comment (RC4) · Anonymous Referee #3 · 4 Nov 2016

The paper deals with the use of satellite-based evapotranspiration estimates (GAET) to improve results of a simple hydrological model. The general idea of the paper is sound and potentially useful for the hydrological community.

Unfortunately, I see a number of problems with the paper. The main problem for me is the unclear rationale of the methodology. GAET is used in two ways to improve the hydrological model, and the two procedures have problems.

The first procedure "constrains" the hydrological model estimates of evapotranspiration HAET forcing them to be more similar to HAET. This is done in a very prescriptive way, and to some extent may contradict the whole physical basis of the model. The results of this exercise are not successful, as shown by the poor performance of the

model in terms of streamflow. There are other ways of constraining intermediate model results, which are more formal and do not compromise the model physics (for example calibration optimization with side constrains). I believe that the first procedure does not present any novelty in terms of ideas or techniques. A thorough justification of why it should be included in the paper based on similar procedures applied successfully elsewhere is needed here.

The second procedure modifies the structure of the model by multiplying the ET equation by a factor. Different formulations are used for the factor, which try to capture more of the physics of the problem. This last point is not clearly explained or justified by the authors. The formulations are tested against GAET estimates and the more complex formulation gives the best results. That formulation is then used to predict discharges, which shows some improvement of the model results. A major problem with the procedure is that the model produces a value of the soil moisture storage capacity H that is totally unrealistic (H=12.8 m). The authors do not report the value of H for the original model without the "improvements" using GAET, but my impression is that it may have been more physically adequate. I think that the author's claim about the advantage of physically-based over data-driven models is weakened by this outcome.

In the middle of all this there are a number of methodological details that are also of concern. For example, model calibration is done using the SCE-UA algorithm, which essentially consists of a global optimization method. Since the formulation of the second procedure involves more calibration parameters, how does that affect the optimization? Also, there are ways of optimizing parameters with constrains that could be explored as a more formal way of incorporating the additional information from the GAET.

Organization is also an issue. There is material in the results that should be in the methods (for example most of 3.1.2. in the results is about how to implement the "constrain" in the model and should be moved to 2.4. study approach). There is also an excessive use of subtitles and dot point type paragraph, which results in a lack of

flow throughout the paper.

One lingering question that I have after reading the paper is why this new methodology was used in a study case with limited data and not on a catchment with extensive data where more verification and checks could be done. After all, the essence of the paper to me is the new formulation to improve an existing hydrological model and from that point of view a better set of data for validation is necessary. I would also add that the application to just one catchment may not be enough to demonstrate that the new formulation is better.

---

## Short Comment (SC5) · 5 Nov 2016

Evapotranspiration (ET) is an extremely important component of catchment water balance. Although calibrated conceptual hydrological models (eg, HYMOD) are able to simulate observed streamflow with relative confidence, past studies have lacked focus in validating another of its output, the model simulated ET. This article utilizes a newly available satellite based ET data over a sparsely gauged catchment in Africa to improve the model structure of HYMOD for a better representation of ET processes. The authors show that the modified model structure not only improves its simulation of ET over the basin by a significant margin, but also improves the overall simulated streamflow.

[Figure]

Improving the structure of a conceptual hydrological model is extremely important for hydrological prediction and forecasting. In a sparsely gauged catchment, observed streamflows are often of poor in quality and are only available for a small number of years. Therefore, instead of only calibrating the model parameters against such data, its also important to incorporate additional information (eg, satellite based ET data) to improve the model structure in the first place. Once the structure is fixed the parameters can be calibrated, as done in this study.

—————————————————————

---

## Author Comment (AC9) · 6 Nov 2016

We thank Mr. Mukherjee for providing his valuable comments on our manuscript. As he rightly pointed out, the primary focus of many conceptual hydrologic models has been in better simulating the streamflow, while not giving much importance to ET. Therefore, the conceptual models should be improved so that they can predict ET with reasonable accuracy. These improved models can then be used for addressing many interesting research questions. For example, we have already started investigating the future water availability in the Mara River basin using the model developed in this study (Hy-ModV2), where ET is an important entity.

We also thank Mr. Mukherjee for pointing out the importance of the structural modifi-

cations in the model.

---

## Author Comment (AC10) · 6 Nov 2016

* * *
[REVIEWER COMMENT 1] The paper deals with the use of satellite-based evapotranspiration estimates (GAET) to improve results of a simple hydrological model. The general idea of the paper is sound and potentially useful for the hydrological community. Unfortunately, I see a number of problems with the paper. The main problem for me is the unclear rationale of the methodology. GAET is used in two ways to improve the hydrological model, and the two procedures have problems.

[AUTHOR COMMENT 1] We thank the reviewer for his/her valuable comments and

acknowledging the importance and relevance of our paper by stating that 'the general idea of the paper is sound and potentially useful for the hydrological community'. We have now thoroughly addressed all of his/her comments and concerns in our response.
* * *
[REVIEWER COMMENT 2] The first procedure "constrains" the hydrological model estimates of evapotranspiration HAET forcing them to be more similar to HAET. This is done in a very prescriptive way, and to some extent may contradict the whole physical basis of the model. The results of this exercise are not successful, as shown by the poor performance of the model in terms of streamflow. There are other ways of constraining intermediate model results, which are more formal and do not compromise the model physics (for example calibration optimization with side constrains). I believe that the first procedure does not present any novelty in terms of ideas or techniques. A thorough justification of why it should be included in the paper based on similar procedures applied successfully elsewhere is needed here.

[AUTHOR COMMENT 2] The reviewer expressed two main concerns in this paragraph. First, the way the method is implemented and second, the performance improvements. We fail to agree with the reviewer on either of them.

Regarding the first point, our constraining scheme is conceptually analogous to any filtering technique, where the main goal is to fix the behavior of the model, not its structure/process parameterization. The model state at any time step is adjusted based on the observation from that time step so that the model behaves more 'accurately'. A filtering cannot directly correct the model structure. Likewise, in our constraining approach, we try to fix the model behavior without modifying its structure. We modify the structure diagnostically in the next step (Step II).

The constraining approach corrects the model behavior in a physically-based manner (using new information from the satellite-based actual ET, GLEAM), which is exactly what we want. The water balance, as expected, is also preserved. Therefore, we don't

agree that the constraining approach contradicts the physical basis of the model. To our opinion, it actually corrects the model behavior.

The reviewer mentions 'calibration optimization', however we are not sure what he meant by that. Calibration itself is optimization. To note, we are already performing calibration (using SCE-UA which is a global optimization algorithm) using two different types of constraints, one on the parameters (their ranges) and the other on the ET process. This should result into a more physically-consistent model and not 'contradict the physical basis of the model'.

We think that the constraining is an important part of the paper and should remain in it.

Regarding the second point, we did have performance improvement in constraining, although not as much as the second approach, where we changed the model structure. In Table 5 (revised manuscript) it can be seen that in many cases the error statistics improve from Step-I to Step-IV. In calibration, NMSE changes from 0.56 to 0.43, NB$\sigma$ changes from -0.12 to -0.06, and R changes from 0.76 to 0.83. In Figure 9 (revised manuscript), we do see improvements in the streamflow simulations (compare the blue and green lines in the transformed space to see the improvements more clearly).

————————————————————————————————————————————————

[REVIEWER COMMENT 3] The second procedure modifies the structure of the model by multiplying the ET equation by a factor. Different formulations are used for the factor, which try to capture more of the physics of the problem. This last point is not clearly explained or justified by the authors. The formulations are tested against GAET estimates and the more complex formulation gives the best results. That formulation is then used to predict discharges, which shows some improvement of the model results. A major problem with the procedure is that the model produces a value of the soil moisture storage capacity H that is totally unrealistic (H=12.8 m). The authors do not report the value of H for the original model without the "improvements" using GAET, but my impression is that it may have been more physically adequate. I think that

the author's claim about the advantage of physically-based over data-driven models is weakened by this outcome.

[AUTHOR COMMENT 3] We thank the reviewer for helping us improve our explanations. As suggested, we have now included the idea (i.e. the formulations 'try to capture more of the physics of the problem') into our discussion.

There has been a misunderstanding here. The reviewer pointed out an H value of 12.8 m, however, our final model DID NOT produce an H value of 12.8 m. Please refer to Appendix C where we provided a table with all the calibrated parameters. The column with the final model in the second step (Step II-1d) shows an H value of 866 mm or 0.866 m, which is conceptually realistic. The value the reviewer mentioned is from another step (Step II-3) which is not the final step. Please note that the H values from all different models (including the original one) are already reported in the table in Appendix C.

Let us try to explain our methodology once again. There is a reason why we designed the study the way it is there in the paper. It is an inclusive design that unifies several different aspects. We have two main steps, Step-I and Step-II. Step-I is based on process constraining, whereas Step-II is based on diagnostic structural modification. In each step, we have five main sub-steps (1 to 5). The first sub-step (Step I-1) is the benchmark (calibrated model WITHOUT constraining or structural modification), the second (Step I-2) is based on imposing the ET constraint on the model ET process without any recalibration (note we still use the calibrated model from the benchmark but we don't recalibrate it), the third (Step I-3) is based on recalibrating the model with ET constraint, the fourth (Step I-4) is based on constraint adjustment, and finally in the fifth (Step I-5) we remove the ET constraint to see how sensitive the performance of the new model is when the satellite ET data become unavailable (note this is no longer the benchmark model since we recalibrated the parameters in Step I-4). We follow the exact same steps in Step II (Step II-1, Step II-2, Step II-3, Step II-4, and Step II-5). The only change here is that we have four more sub-steps (a-d) in Step II-1. These

are based on the different structures we used. We select the best structure from Step II-1 and then carry out the other steps to see if there is any benefit in constraining the modified model itself. Our results indicate that there isn't much benefit in constraining the modified model and so we select the model with the best structure (without constraining) from Step II-1. Thus, our final model is the one from Step II-1d.
* * *
[REVIEWER COMMENT 4] In the middle of all this there are a number of methodological details that are also of concern. For example, model calibration is done using the SCE-UA algorithm, which essentially consists of a global optimization method. Since the formulation of the second procedure involves more calibration parameters, how does that affect the optimization?

[AUTHOR COMMENT 4] All the calibration runs were carried out with the same settings of SCE-UA, i.e. with the same number of complex and loops, in order to nullify the effects of the optimization algorithm itself. The calibration runs were successful in all the cases. We have already reported the calibrated parameters from all different calibration runs in the appendix.
* * *
[REVIEWER COMMENT 5] Also, there are ways of optimizing parameters with constrains that could be explored as a more formal way of incorporating the additional information from the GAET.

[AUTHOR COMMENT 5] We have already addressed the issue of calibration with constraints in one of our previous paragraphs (see Author Comment 2). The constraint could be on the parameters or it could be on the processes. We are already applying constraints on the parameters by setting their limits. We are also imposing constraints on the ET process within the model using the satellite ET data. Regarding constraining with ET, we found two very good papers (Winsemius et al., 2008, van Emmerik et al.,

2015, already cited in our paper) where the authors constrain the model parameters sensitive to ET using the ET data. Note that in this study our approach (and goal) is different. We impose the constraint on the ET process and also modify the model structure.
* * *
[REVIEWER COMMENT 6] Organization is also an issue. There is material in the results that should be in the methods (for example most of 3.1.2. in the results is about how to implement the "constrain" in the model and should be moved to 2.4. study approach). There is also an excessive use of subtitles and dot point type paragraph, which results in a lack of flow throughout the paper.

[AUTHOR COMMENT 6] We thank the reviewer for pointing this out. This was also pointed out by Reviewer #2. We have already taken care of this issue in the revised manuscript (also uploaded on the discussion forum).
* * *
[REVIEWER COMMENT 7] One lingering question that I have after reading the paper is why this new methodology was used in a study case with limited data and not on a catchment with extensive data where more verification and checks could be done. After all, the essence of the paper to me is the new formulation to improve an existing hydrological model and from that point of view a better set of data for validation is necessary. I would also add that the application to just one catchment may not be enough to demonstrate that the new formulation is better.

[AUTHOR COMMENT 7] Please note that one of the main purposes of this study is to develop models for sparsely-gauged basins. That is why we are using the satellite-based actual ET data in the first place. In a well instrumented catchment we could instead use flux tower data directly. A method/model that has worked well in a highly-instrumented catchment doesn't necessarily guarantee that it will also work well in a

sparsely-gauged catchment.

The main focus of our project (NASA SERVIR) has been in solving water resources problems in sparsely-gauged basins using observations from space, and our current study is well aligned with the objective and the scope of the project.

We understand the usefulness of testing a model/method in multiple catchments (see the paper Gupta et al. (2014, HESS) by one of the coauthors of this paper), however, doing that was beyond the scope of this study. A rigorous testing is one of our future plans.

Please refer to the paragraph in '4.3 Overall Outlook' where we discuss this issue:

'Note that this study is based on testing of a single catchment scale conceptual rainfall-runoff model on a single basin, using a single satellite-based precipitation product and a single satellite-based AET product. While not demonstrating universal applicability, the results are clearly indicative and the methodology illustrates how such data can be used to investigate potential improvements to the structures of simple catchment scale models used for hydrologic studies in data scarce regions. For more detailed process-based models, the ET process parameters can be calibrated against some reliable SET estimates (e.g. GLEAM), or the process representation itself can be improved by adapting some similar strategies the SET products are based on.'
* * *
References

van Emmerik, T., Mulder, G., Eilander, D., Piet, M. and Savenije, H.: Predicting the un-gauged basin: model validation and realism assessment, Front. Earth Sci., 3(October), 1–11, doi:10.3389/feart.2015.00062, 2015.

Gupta, H. V., Perrin, C., Blöschl, G., Montanari, A., Kumar, R., Clark, M. and Andréassian, V.: Large-sample hydrology: a need to balance depth with breadth, Hydrol. Earth Syst. Sci., 18(2), 463–477, doi:10.5194/hess-18-463-2014, 2014.

Winsemius, H. C., Savenije, H. H. G. and Bastiaanssen, W. G. M.: Constraining model parameters on remotely sensed evaporation: justification for distribution in ungauged basins?, Hydrol. Earth Syst. Sci., 5(4), 2293–2318, doi:10.5194/hessd-5-2293-2008, 2008.

Please also note the supplement to this comment:
http://www.hydrol-earth-syst-sci-discuss.net/hess-2016-413/hess-2016-413-AC10-supplement.pdf
* * *

---

## Author Response (AR1)

**CONTENT**
* * *
**IMPORTANT POINTS**

1. All of the reviewers' comments are now addressed.
2. All of the editor's comments are now addressed.
3. Section 2 (Study Area, Data and Methodology), Section 3 (Results), and Section 4 (Discussion) are extensively rewritten in the revised manuscript to take care of the concerns expressed by the reviewers and the editor.
4. Two new figures (Fig. 4 and 5) are now added to the revised manuscript, one explaining the updated model and the other describing the methodology.
5. Previously uploaded review responses are updated according to the revised manuscript, as necessary.
* * *
**RESPONSE TO THE EDITOR**

**Editor:**

**We have now received the reports of three referees and the responses of the authors including a revised manuscript. My general view is that the paper is technically sound and the contents are relevant for the audience of HESS. However, I found the paper hard to read (particularly the methodology part that is very hard to follow) as did reviewers 2 and 3.**

**Authors:**

Dear Editor,

Thank you for your constructive comments, which have helped us to improve and revise our manuscript. We believe that the readability of the manuscript has been significantly improved as a result. Please also see our response to your specific comments in the following. Please also note that we have made some (minor) updates to our previously uploaded review responses according to the revised manuscript.

The review comments really helped us a lot in improving the manuscript and we thank the reviewers for that. Also, many thanks to the commenters for their constructive inputs.

**Editor:**

**There are a number of issues that need consideration, including a better organization and synthesis and a more detailed explanation of methods and physical implications. These issues are listed in the comments by the reviewers, and though I realize that they have been partially addressed by the authors, the revised manuscript still needs further work in this regards.**

**Authors:**

In the revised manuscript, we have made significant changes to the results and discussion sections. Please refer to the version with tracked changes to see the changes made.

**Editor:**

**The study approach section needs considerable reworking. The dot point format is hard to follow and includes few details. Some of the material here should include explanations that are in the results section like, for example, how constrains are implemented. Other aspects that need more explanation are the calibration/optimisation procedure and the determination of parameters of the new formulation.**

**Authors:**

We have now avoided the dot point format in our revised manuscript. We have also transferred a lot of contents from the results section to the methods section. Both sections are now extensively rewritten. We have also included some more details about the calibration/optimization procedures. There is a new figure (Fig. 5) added to the method section demonstrating our study approach.

**Editor:**

**There are also too many subtitles, the authors should make an effort to provide a section with a better flow and easier to read.**

**Authors:**

We have now removed most of the sub-titles and made the sections easier to read by maintaining a flow.

**Editor:**

**There is also not enough discussion on the physical implications of the newly improved model HyMod V2. Figure 4 is a start but more can be said in the text, particularly in the results or discussion sections.**

**Authors:**

We have now expanded the discussion on the newly improved model.

**Editor:**

**Please note that, after the revision stage, the revised manuscript and author responses will be sent to the referees to complete the review process.**

**Authors:**

Thank you for this information.
* * *
**RESPONSE TO REVIEWER #1**

**Authors:**

We thank the reviewer, who identified himself as Prof. Abdolreza Bahremand, for nicely summarizing the key aspects of our study and pointing out their importance. We have addressed all of his comments and made the suggested changes in our revised manuscript.

NOTE: Page and line numbers mentioned in the response correspond to the revised manuscript.

**Reviewer:**

**The paper and its discussion and conclusion has much more other useful contents than what has been given in the abstract. I guess the limitation of the abstract word numbers (500 words) has been the reason for this. 2. One concluding sentence (like those written in the conclusion) should be added here.**

**Authors:**

As per the reviewer's suggestion, we have now modified the last sentence of the abstract as in the following to summarize our main outlook:

"*Results indicate that while both approaches can provide improved simulations of streamflow, the second approach significantly improves the simulation of actual evapotranspiration, which substantiates the importance of making 'diagnostic structural improvements' to hydrologic models whenever possible.*"

**Reviewer:**

**GLEAM and HyMod could be other keywords for this paper? Don't you think so?**

**Authors:**

We agree with the reviewer on this and have now included both GLEAM and HyMod as keywords.

**Reviewer:**

**Here, in such case I am sure the authors know better than me that the strong correlation is not enough :)**

**Authors:**

Yes, we agree that a strong correlation is not enough. Therefore, in our revised manuscript, we have now included a detailed discussion on the evaluation of GLEAM using several other error statistics. We are now citing one book chapter and four papers for this discussion.

Page 2 Line 21 – Page 3 Line 4

"... *Worldwide evaluations suggest that satellite-based ET estimates are strongly correlated (~0.83) with ground-based observations made at flux towers (Demaria and Serrat-Capdevila, 2016).*

*For this study, we use the Global Land Evaporation Amsterdam Model (GLEAM) as the source of the satellite-based ET (SET) data. In the GLEAM algorithm, ET is computed using only a small number of satellite-based inputs, which makes it particularly beneficial for application to sparsely gauged basins. Miralles et al. (2011) have shown that GLEAM estimates of evaporation are strongly correlated (0.80) with annual cumulative evaporation estimated via eddy covariance at 43 stations, and have very low (-5%) average bias. The correlations at individual stations are strong (0.83) for all vegetation and climate conditions, and improve to 0.9 for monthly time series (Miralles et al., 2011). McCabe et al. (2016) reported satisfactory statistical performance (R2 = 0.68; Root Mean Square Difference = 64 Wm–2; Nash-Sutcliffe Efficiency = 0.62) of GLEAM when compared against data from 45 globally-distributed eddy-covariance stations. Michel et al. (2016) compared Priestley-Taylor Jet Propulsion Laboratory model (PT-JPL), Moderate Resolution Imaging Spectroradiometer evaporation product (PM-MOD), Surface Energy Balance System (SEBS), and GLEAM simulations against 22 FLUXNET tower-based flux observations and found GLEAM and PT-JPL to more closely match in-situ observations for the selected towers and reference period (2005-2007). Their extended analysis over 85 towers also had a similar overall outcome. Miralles et al. (2016) compared three process-based ET methods (PM-MOD, GLEAM and PT-JPL) against surface water balance from 837 globally distributed catchments, and reported that GLEAM and PT-JPL provide more realistic estimates of ET. They found these two products to provide superior overall performance for most ecosystem and climate regimes, whereas PM-MOD tends to underestimate the flux in tropics and subtropics.*"

**Reviewer:**

**and whether the model provides improved ...........[here, i proposed this little correction to make it in harmony with the previous sentences. So, i think adding the conjunction "whether", like what you did for the previous sentence, is better here.]**

**Authors:**

Thanks for the suggestion. We have now modified the sentence as:

Page 3 Line 16-17

"*Finally, we test whether the use of GLEAM SET can further improve the performance of the structurally modified model, and whether there is any decline in model performance if GLEAM SET data become unavailable.*"

**Reviewer:**

**Unnecessary abbreviation makes the text a bit boring, as the text has already too many :) If you wish to reduce them then just start with the name of the rivers...**

**Authors:**

We agree with the reviewer on this. We have now removed the unnecessary abbreviations from our revised manuscript.

**Reviewer:**

**The abbreviation, HAET, has not been introduced in the paper so far. Perhaps you mean HyMod AET. First I thought that H stands for Hargreaves...**

**Authors:**

Thanks for pointing this out. We have now introduced HAET in Section 2.3.

**Reviewer:**

**after the remove of the GAET data.**

**Authors:**

We have now modified this part of the sentence as:

"*... after the removal of the GAET data.*"

**Reviewer:**

**This sentence needs a little bit of improvement (grammar correction).**

**Authors:**

Thanks for pointing this out. We changed the sentence as in the following:

"Therefore, our results suggest that ET constraining approach should be implemented only for the simulation periods when SET data are available."

However, this statement seemed redundant once the constraining results were already discussed. Therefore, we deleted this from the revised manuscript.
* * *
RESPONSE TO REVIEWER #2

**Authors:**

We thank the reviewer for reviewing our manuscript and providing his/her valuable feedbacks. We have now addressed all of his/her comments and discussed them in the following. As the reviewer mentioned, there were some places in the manuscript which created confusions and the concepts seemed circular. We agree with the reviewer on that. These were mainly due to the lack of sufficient care in the use of terminology. We have revised the manuscript to resolve these issues and make our message more clear-cut. Thanks to the reviewer's feedback, the paper is now much improved.

**Reviewer:**

**This paper presents results from a study examining the use of satellite estimates of actual evapotranspiration (SET) to firstly constrain and secondly modify a HyMod model of Nyangores River Basin in Kenya. Although the ideas presented here are interesting, I found that the reasoning used in the study was circular and I'm not convinced by the results. I think the presentation of the material is too much like a report and the method and results are often mixed up, with the vast majority of the method discussion provided in Section 3 which is nominally the results section. The paper also refers to another publication in preparation by the same authors on this catchment and without seeing this it is difficult to understand the similarity and any potential overlaps between the two publications. It's not clear why this paper would be presented first. I recommend that the paper is rejected and the authors undertake more extensive validation of the method in a catchment where there is data other than the SET to allow comparisons.**

**Authors:**

The manuscript is designed such that all the analyses steps are clearly stated and their results are thoroughly discussed. This is important since we recommend this approach for similar investigations, due to the fact that it's inclusive. It takes into account several important issues, including process constraining, use of constraint adjustment, usefulness of model (re)calibration, utilizing new information (from satellite-based sources), diagnostic model structural improvement, and uncertainty analysis. However, as pointed out by the reviewer, we do see that some method discussions could be removed from the

results section and put back to the methods section itself. We have now made significant changes in both of these sections to take care of this issue in our revised manuscript.

Regarding the point on the second publication, we do have another manuscript under review, however, we would like to clarify that the objective and scope of that manuscript are quite different as compared to this one. That manuscript reports on the development of a multi-model and multi-product (satellite)-based probabilistic operational streamflow forecasting platform for sparsely-gauged basins and does not in any way address the problem of model structural correction/improvement. We are ready to share the manuscript with the reviewer and the editor personally if necessary to resolve this concern.

Since the other manuscript is under review, we are not citing that anymore in this manuscript.

Regarding the comment on other available data for comparison, note that the dataset (GLEAM) we are using has already been validated in several recent studies. Although we didn't include the detailed discussion on validation in our initial manuscript, we have now included that part in our revised manuscript (Page 2 Line 21 - Page 3 Line 4). GLEAM has already been evaluated both at local (eddy covariance towers) and global scales. There have been projects that have focused on the topic of the evaluation of GLEAM, e.g. WAter Cycle Multi-mission Observation Strategy-EvapoTranspiration (WACMOS-ET), Global Energy and Water Cycle Exchanges (GEWEX) LandFlux Project, etc.

Several studies (also cited in our manuscript) have found GLEAM to be one of the best ET products. Therefore, we don't think it is necessary to carry out an additional evaluation of GLEAM, given the fact that other studies have already focused on that part. This also does not fit well with the main goals of this manuscript. Moreover, an evaluation study of this kind would stand out on its own as an independent paper, which is clearly beyond the scope of this manuscript.

To be clear, the main objective of this study is NOT to validate/compare actual ET products, which is an interesting topic, but appropriate for a different manuscript. In this study, we explore different structure-related methods (including process constraining) to improve the performance of a rainfall-runoff model, and we have an inclusive design to organize all the steps in a systematic manner. We show how the model deficiencies could be overcome by using new sources of information

**Reviewer:**

**If I understand the method properly, in Case 1 HyMod is run and the AET from the model is found to be different from the SET estimates. So the model is run using SET to constrain the AET in the model by setting the requirement that the AET <= SET. However then the model parameters are found to be unrealistic so the SET is bias corrected so that when the model is constrained to have AET <= SET, the model parameters are more realistic. In all of this there is no evaluation of the SET itself**

**and the bias correction step implies that there are problems with the SET. So you're trying to match a model to a biased quantity and then changing that quantity and then still trying to match it. It just seems very circular to me. Case 2 follows much the same logic except rather that using the constraint that AET <= SET, the model structure is changed with a variety of different equations that factor the evaporative demand ratio. Finally in Figure 9 the model is compared back to the SET which was used to correct the model I just don't understand how you can accept the SET data without having an external validation. I accept that this is unlikely to exist for the catchment you have chosen but I think you then need to test your method in a more instrumented catchment where you do have external validation data and once you have confidence in the method then you can apply to a poorly gauged basin.**

**Authors:**

This is an important point which we unfortunately did not explain well in the original manuscript. We thank the reviewer for pointing this out. Note that we are not doing any bias-correction of GLEAM in this study, because for that we needed the 'ground truth' of the observed ET. Furthermore, GLEAM has been validated in several recent studies (already cited in our manuscript). Therefore, the term 'bias-correction' was wrongly used and we have now changed that. A more appropriate term in this case would be 'constraint adjustment', which is what we are essentially doing. In Stage-I, the model structure is fixed. When GAET is used as a constraint to the ET process within the model, it introduces bias in the streamflows. Therefore, we adjust the constraint such that that bias is removed. Note that this is NOT indicative of the presence of any actual bias within the GAET estimates. The constraint adjustment factor is a model 'parameter' which corresponds to the structural deficiencies within the model. It may or may not be necessary as the structure changes. In Stage-II analysis, we saw that when the structure was improved (deficiencies reduced), ET constraint adjustment became irrelevant.

Regarding the point of external validation, please see the last three paragraphs of our first response (to Reviewer #2).

**Reviewer:**

**Page 2 – paragraph 3 – at this stage its not clear how ET can be a model target – I think you need to make it clearer at this point that PET is forcing data and AET is a model state.**

**Authors:**

We consider precipitation and PET as the forcings. Note that the precipitation is the only input to the water budget of the model, PET is a constraint to set the upper limit of the actual ET in the original HyMod model. The model produces both discharge and actual ET as outputs. Therefore, we don't see why AET needs to be considered as a model state (as conventionally defined). It is a model simulated output.

**Reviewer:**

**Page 2, line 15 – good correlation of the SET does not give me confidence that the property is not biased which is key for this method and even line 23 where the annual bias is low doesn't guarantee that there are not other biases that are cancelling out throughout the year.**

**Authors:**

We agree with the reviewer on the point of the value of correlation. Actually we had a very brief discussion on the comparison/validation of the ET products in our initial manuscript. We have now expanded that discussion in our revised manuscript (Page 2 Line 21 – Page 3 Line 4), where some additional error statistics (apart from correlation coefficient) are also reported.

Note this was also pointed out by Reviewer #1 and in reply to his comments we showed the new paragraph that has been added to the revised manuscript (Page 4 of this document).

**Reviewer:**

**Page 4 – paragraph 12 – TRMM data is no longer available so not clear why you say that it is available to near-present? The study period is not clear from Section 2 in any case.**

**Authors:**

This is an important point. We have now included this information in our revised manuscript. We are using the TRMM Multi-Satellite Precipitation Analysis (TMPA-RT) dataset which is still available. This is a merged dataset. TRMM Microwave Imager (TMI) was a part of it, which is no more operational (since 8 April 2015) because of fuel and battery issues with the satellite. As mentioned by the developers, the absence of TRMM is not crucial to the production of TMPA and TMPA-RT data.

We discuss the time periods in Section 2.5:

"*The model was run continuously for the 7.5-year period Jan 2003 to June 2010, with the first 4 years (2003 to 2006) used for calibration and the remaining 3.5 years (2007 to mid-2010) used to provide an additional assessment of model performance. Results are shown for the "calibration (4-years)", "evaluation (3.5 years)" and "total (7.5 years)" simulation periods.*"

**Reviewer:**

**Page 4, line 34 – here you describe Stage 1 as "constraining" and you are at pains to point out that it is not assimilation and yet in the remainder of the manuscript you continue to use the term assimilation – I think you need to be more careful with the terminology e.g. Page 8, line 23; Page 12, Line 24**

**Authors:**

Thank you for pointing this out. We have now removed the term 'assimilation' wherever required.

**Reviewer:**

**Page 5, Step 1-2 – given this is the method section, there are no details here of the actual constraints. These are provided in the results section. I think this makes the presentation quite confused and doesn't provide the reader with much of a sign post or guide as where the research is heading. Similar comments for Step II-1 where the four equations are mentioned.**

**Authors:**

We agree with the reviewer on this, and we want to point out that we have now made significant changes in the methods, results, and discussion sections in our revised manuscript to take care of this issue.

**Reviewer:**

**Page 7, Line 24 – I don't understand why you validate your water balance using satellite precipitation which has its own concerns. Why not use some ground based data as well?**

**Authors:**

Note that the TMPA data used in this study has been bias corrected using rain gauge measurements from the study area. The detailed methodology is discussed in the other manuscript (streamflow forecasting). As mentioned earlier, we are ready to share the manuscript personally with the reviewer or the editor.

**Reviewer:**

**Page 7, Line 27 – "based on our expectation of how it would behave" – this comes to my concern about the validation. We generally expect a more robust validation than just a sense that the soil moisture should be smooth. Why should it be smooth for this catchment? You don't appear to have any soil moisture data to validate this statement.**

**Authors:**

Thanks for pointing this out. We agree that in order to make this statement, the soil moisture data need to be studied first. Therefore, we have now removed this sentence from our revised manuscript.

**RESPONSE TO REVIEWER #3**

**Reviewer:**

**The paper deals with the use of satellite-based evapotranspiration estimates (GAET) to improve results of a simple hydrological model. The general idea of the paper is sound and potentially useful for the hydrological community.**

**Unfortunately, I see a number of problems with the paper. The main problem for me is the unclear rationale of the methodology. GAET is used in two ways to improve the hydrological model, and the two procedures have problems.**

**Authors:**

We thank the reviewer for his/her valuable comments and acknowledging the importance and relevance of our paper by stating that 'the general idea of the paper is sound and potentially useful for the hydrological community'. We have now thoroughly addressed all of his/her comments and concerns in our response in the following.

**Reviewer:**

**The first procedure "constrains" the hydrological model estimates of evapotranspiration HAET forcing them to be more similar to HAET. This is done in a very prescriptive way, and to some extent may contradict the whole physical basis of the model. The results of this exercise are not successful, as shown by the poor performance of the model in terms of streamflow. There are other ways of constraining intermediate model results, which are more formal and do not compromise the model physics (for example calibration optimization with side constrains). I believe that the first procedure does not present any novelty in terms of ideas or techniques. A thorough justification of why it should be included in the paper based on similar procedures applied successfully elsewhere is needed here.**

**Authors:**

The reviewer expressed two main concerns in this paragraph. First, the way the method is implemented and second, the performance improvements. We fail to agree with the reviewer on either of them.

Regarding the first point, our constraining scheme is conceptually analogous to any filtering technique, where the main goal is to fix the behavior of the model, not its structure/process parameterization. In filtering, the model state at any time step is adjusted based on the observation from that time step so that the model behaves 'more accurately'. A filtering cannot and is not meant to directly correct the model structure. Likewise, in our constraining approach, we try to fix the model behavior without modifying its structure. We modify the structure diagnostically in the next step (Stage-II).

The constraining approach corrects the model behavior in a physically-consistent manner (using new information from the satellite-based actual ET, GLEAM), which is exactly what we want. The water balance, as expected, is also preserved. Therefore, we don't agree that the constraining approach contradicts the physical basis of the model. To our opinion, it actually corrects the model behavior.

Regarding 'calibration optimization with side constraints', note that we are already performing calibration (using SCE-UA which is a global optimization algorithm) using two different types of constraints, one on the parameters (their ranges) and the other on the ET process. This should result into a more physically-consistent model and not 'contradict the physical basis of the model'.

We think that the constraining is an important part of the paper and should remain in it.

Regarding the second point, we did have performance improvement in constraining, although not as much as the second approach, where we changed the model structure. In Table 5 (revised manuscript) it can be seen that in many cases the error statistics improve from Step-1 to Step-4. In calibration, NMSE changes from 0.56 to 0.43, NBσ changes from -0.12 to -0.06, and R changes from 0.76 to 0.83. In Figure 10 (revised manuscript), we do see improvements in the streamflow simulations (compare the blue and green lines in the transformed space to see the improvements more clearly).

**Reviewer:**

**The second procedure modifies the structure of the model by multiplying the ET equation by a factor. Different formulations are used for the factor, which try to capture more of the physics of the problem. This last point is not clearly explained or justified by the authors. The formulations are tested against GAET estimates and the more complex formulation gives the best results. That formulation is then used to predict discharges, which shows some improvement of the model results. A major problem with the procedure is that the model produces a value of the soil moisture storage capacity H that is totally unrealistic (H=12.8 m). The authors do not report**

**the value of H for the original model without the "improvements" using GAET, but my impression is that it may have been more physically adequate. I think that the author's claim about the advantage of physically-based over data-driven models is weakened by this outcome.**

**Authors:**

We thank the reviewer for helping us improve our explanations. As suggested, we have now explained the idea (i.e. the formulations 'try to capture more of the physics of the problem') more clearly in our revised manuscript.

Please note that there has been a misunderstanding here. The reviewer pointed out an H value of 12.8 m, however, our final model DID NOT produce an H value of 12.8 m. Please refer to Appendix C where we provided a table with all the calibrated parameters. The column with the final model (Stage-II Step-1 Case-D) shows an H value of 866 mm or 0.866 m, which is conceptually realistic. The value the reviewer mentioned is from another step (Stage-II Step-3) which is not the final step. Please note that the H values from all different models (including the original one) are already reported in the table in Appendix C.

We understand that one of the reasons this confusion arose was because our methods and results sections were not written properly in the initial manuscript. In our revised manuscript, we have made significant changes in these sections to present the content more clearly. Hopefully this will guide the readers much better.

Following is a paragraph from the revised manuscript explaining our methodology:

"*The entire study approach is summarized in Fig. 5. As can be seen, only Step-1 is different for both the stages (Stage-I and Stage-II), while the remaining four steps (Step 2-5) are similar. Thus, in each stage, there are five steps altogether. Stage-I Step-1 is for generating benchmark simulations using the calibrated model but without any ET constraint or structural modifications. On the other hand, Stage-II Step-I has four different cases (A-D) corresponding to different structural modifications in the ET process parameterization. Both the benchmark model from Stage-I Step-I and the best performing model from Stage-II Step-I are used in the following steps. Step-2 is based on imposing the ET constraint but without recalibration, meaning that the same set of calibrated parameters as in the benchmark step is further used in this step. Step-3 is based on recalibrating the model while imposing the ET constraint. Step-4 is conceptually similar to Step-3, however, additionally, some constraint adjustments (Eq. 1 and 2) are applied and the adjustment parameters are calibrated together with model parameters (to match the simulated and observed streamflows). Finally, in Step-5, we remove the ET constraint to see whether the performance of the new model will decline when satellite ET data becomes unavailable (note this is no longer the benchmark model since we recalibrated the parameters in Step 4).*"

**Reviewer:**

**In the middle of all this there are a number of methodological details that are also of concern. For example, model calibration is done using the SCE-UA algorithm, which essentially consists of a global optimization method. Since the formulation of the second procedure involves more calibration parameters, how does that affect the optimization?**

**Authors:**

All the calibration runs were carried out with the same settings of SCE-UA, i.e. with the same number of complex and loops, in order to nullify the effects of the optimization algorithm itself. The calibration runs were successful in all the cases. We have already reported the calibrated parameters from all different calibration runs in the appendix.

**Reviewer:**

**Also, there are ways of optimizing parameters with constrains that could be explored as a more formal way of incorporating the additional information from the GAET.**

**Authors:**

We have already addressed the issue of calibration with constraints in one of our previous paragraphs (response to Reviewer #3). The constraint could be on the parameters or it could be on the processes. We are applying constraints on the parameters by setting their limits. We are also imposing constraints on the ET process within the model using the satellite ET data.

Regarding constraining with ET, we found two very good papers (Winsemius et al., 2008, van Emmerik et al., 2015, already cited in our paper) where the authors constrain the model parameters sensitive to ET using the ET data. Note that in this study our approach (and goal) is different. We impose the constraint on the ET process and also modify the model structure.

**Reviewer:**

**Organization is also an issue. There is material in the results that should be in the methods (for example most of 3.1.2. in the results is about how to implement the "constrain" in the model and should be moved to 2.4. study approach). There is also an excessive use of subtitles and dot point type paragraph, which results in a lack of flow throughout the paper.**

**Authors:**

We thank the reviewer for pointing this out. This was also pointed out by Reviewer #2. We have now made several changes in the sections dealing with methodology, results, and discussion, to take care of this issue.

**Reviewer:**

**One lingering question that I have after reading the paper is why this new methodology was used in a study case with limited data and not on a catchment with extensive data where more verification and checks could be done. After all, the essence of the paper to me is the new formulation to improve an existing hydrological model and from that point of view a better set of data for validation is necessary. I would also add that the application to just one catchment may not be enough to demonstrate that the new formulation is better.**

**Authors:**

Please note that one of the main purposes of this study is to develop models for sparsely-gauged basins. That is why we are using the satellite-based actual ET data in the first place. In a well instrumented catchment we could instead use flux tower data directly. A method/model that has worked well in a highly-instrumented catchment doesn't necessarily guarantee that it will also work well in a sparsely-gauged catchment.

The main focus of our project (NASA SERVIR) has been in solving water resources problems in sparsely-gauged basins using observations from space, and our current study is well aligned with the objective and the scope of the project.

We understand the usefulness of testing a model/method in multiple catchments (see the paper Gupta et al. (2014, HESS) by one of the coauthors of this paper), however, doing that was beyond the scope of this study. A rigorous testing is one of our future plans.

Please refer to the last paragraph in 'Discussion' where we point out this issue:

"*Note that this study is based on testing the model on a single basin using a single satellite-based AET product. While not demonstrating universal applicability, the results are clearly indicative and the methodology illustrates how such data can be used to investigate potential improvements to the structures of simple catchment scale models used for hydrologic studies in data scarce regions. A rigorous analyses of the methodology over multiple basins is a potential avenue for future research scope. For more detailed process-based models, the ET process parameters can be calibrated against some reliable satellite-based AET estimates (e.g. GLEAM), or the process representation itself can be improved by adapting some similar strategies that these AET products follow.*"

[revised manuscript text omitted]

[Insert Fig 10]

**4 Discussion and Conclusions**

*[47]* This In this study has, we have explored two different approaches tofor using the use of recently available SET data fromsatellite-based AET dataset GLEAM to improve the realism and performance of the conceptual catchment-scale hydrologic model HyMod. In the first approach, SET data were GAET is used as a constraint to constrain the ET estimatesprocess equation in the model, while in the second we modifiedapproach, the model structure itself. Our study showshas been modified so that use of satellite-based information the ET process parameterizations become more physically consistent and realistic. We avoided making the model overly complicated in terms of its structural representations and/or have large number of

parameters, since both of these would defeat its main purpose of being a simple model. Our goal was to increase the realism within the model and improve its performance in simple manners. Furthermore, we also made sure that the improvements in some particular process simulation (e.g. AET) do not deteriorate model's performance for simulating some other process (e.g. streamflow). Our results show that both the approaches (process constraining and structural modification) can clearlyimprove the simulations of streamflow, while the later also significantly improves the AET simulations. Clearly, the satellite-based ET datasets (GLEAM in this case) can significantly benefit the process of hydrologic modeling forin poorly gauged basins by providing new sources of information to reduce the epistemic component of model structural uncertainty through improved physical process representation.

**4.1 Constraining ET**

*[48]* Use The use of ET data to constrain model simulationsas a constraint can improve streamflow forecasts, provided some additional processing steps are implemented. Direct insertion of If the GAET intodata are used directly as a constraint to the ET equation of, the HyMod model resulted intends to show bias (Step I-2); the in streamflow simulations. This behavior can be attributed to the fact that once GAET is incorporated, the water balance within the model is altered, the effects of which are reflected in terms of bias in the simulated streamflows. The type of this bias will, of course, beis subject to change depending on the SET data used.dataset, While recalibration of the model improved modelwith the ET constraint improves the performance (Step I-3)., it resultedcan result 
[revised manuscript text omitted]

---

## Author Response (AR2)

**AUTHOR COMMENTS**

**01/20/2017**

We are thankful to the editor and the reviewers for their constructive comments and suggestions that helped us a lot to improve our manuscript. Please see our response to the comments regarding the technical corrections.

**Response to the Editor:**

Editor: "... this statement can be further highlighted and strengthened by just slightly changing the wording (as a "need" not just a "potential avenue for research")."

Reply: Thank you for pointing this out. We have now made this change in our manuscript.

**Response to Reviewer #1:**

1. The four cases (A, B, C, and D) are given in capital letters everywhere except in the first table. So, you may please capitalize them in Table 1 too.

Reply. Thank you for pointing this out. We have now fixed this issue.

2. BE is not given in the table 5, while in the appendix A says so! It has been given in Table 1 and the last table located in the appendix B, but not the table 5. So, please correct this in Appendix A, line 19.

Reply: We have now corrected this.

3. In Fig 6, the scatter plot, the axes should be exactly the same. If possible then them identical please. Although some would say that beauty is not only in symmetry, but I would say so :)

Reply: We agree. We have now fixed this issue.

4. In page 8 in the last paragraph, you have written about CSMA. You have introduced this variable as C in the appendix A which is understandable. But you had better mention the CSMA in the appendix A as well (in the appendix, we have C, Cint, Cbeg and Cmax too). You can do this in the line 6 of the appendix in the parenthesis given for the state variable or wherever you would think it is possible.

Reply: Thank you for mentioning this point. To avoid confusions, we have now replaced "$C_{SMA}$" with "C" in Page 8.

5. Page 15, in the first line, in the parenthesis, it should be Case-D instead of Case-C.

Reply: That is right. We have now fixed this issue.

6. Page 21, in line 34, there is a repetition of the word "improve" in one sentence, you may use another word for the second usage of "improve".

Reply: We have now replaced the first "improve" with "adjust".